

# Real-space approach for the Euler class and fragile topology in quasicrystals and amorphous lattices

Dexin Li[1], Citian Wang[1] and Huaqing Huang[1,2,3]⋆

**1** School of Physics, Peking University, Beijing 100871, China
**2** Collaborative Innovation Center of Quantum Matter, Beijing 100871, China
**3** Center for High Energy Physics, Peking University, Beijing 100871, China

⋆ huaqing.huang@pku.edu.cn

## Abstract

We propose a real-space formalism of the topological Euler class, which characterizes the fragile topology of two-dimensional systems with real wave functions. This real-space description is characterized by local Euler markers whose macroscopic average coincides with the Euler number, and it applies equally well to periodic and open boundary conditions for both crystals and noncrystalline systems. We validate this by diagnosing topological phase transitions in clean and disordered crystalline systems with the reality endowed by the space-time inversion symmetry $\mathcal{I}_{ST}$. Furthermore, we demonstrated the topological Euler phases in quasicrystals and even in amorphous lattices lacking any spatial symmetries. Our work not only provides a local characterization of the fragile topology but also significantly extends its territory beyond $\mathcal{I}_{ST}$-symmetric crystalline materials.



# 1  Introduction

Topological phases have garnered attention for their unique properties, originating with the integer quantum Hall effect which is characterized by the topological invariant called the Chern number [1–3] and associated chiral edge modes [4, 5]. Mathematically, the Chern number is derived from the Chern class, a cohomology class characterizing *complex* vector bundles. Typically, Chern numbers can be determined from complex Bloch wave functions via a momentum-space expression that relies on the translation invariance of crystalline solids [6–9]. However, in open-boundary systems, or in the presence of disorder, the lack of transitional invariance renders the momentum-space expression no longer available. This has led to the development of a real-space representation of the Chern number [10], including local Chern markers [11, 12] and the nonlocal Bott index [13, 14], which triggers extensive study on the real-space characterizations of more topological states of matter [15–41].

Recently, novel topological phases characterized by Euler and Stiefel-Whitney classes have been proposed in orientable *real* vector bundles associated with real Bloch states [42–47]. Physically, two-dimensional real wave functions can be topologically classified by the Stiefel-Whitney numbers [48–50] which are $\mathbb{Z}_2$ invariants taking either 0 or 1, and each two-band subspace may exhibit a fragile topology that is characterized by an integer Euler number $e \in \mathbb{Z}$ [51–53]. Similar to the Chern number, the Euler number can be expressed as an integral

in momentum space for real orientable two-band subsystems, and its parity is identical to the second Stiefel-Whitney number $w_2$, implying a close relationship between these two classes. Unlike the Chern insulator, the fragile topology of the Euler class can be tuned by adding trivial bands, implying its non-additive feature [49, 50]. Nevertheless, such a fragile topology protects the nonzero superfluid weight in twisted bilayer graphene [54]. Moreover, the Euler class also serves as non-Abelian topological invariants to characterize the band nodal braiding in multi-gap systems [55], which is in stark contrast to the single-gap Abelian topology within the ten-fold way classification [56]. Such multi-gap non-Abelian topology has been implemented in various systems such as crystalline materials [47, 57], acoustic metamaterials [58–60], and photonic systems [61–64], stimulating rapid recent progress in this ever-growing field [65–68].

Typically, the real Bloch states in crystals are enforced by the space-time inversion symmetry $\mathcal{I}_{ST}$ (time-reversal $\mathcal{T}$ combined with inversion $\mathcal{P}$ or two-fold rotation $\mathcal{C}_{2z}$) [69], which can be destroyed locally in the presence of disorder. Moreover, in a finite nonmagnetic system with open boundaries, $\mathcal{I}_{ST}$ symmetry is not even essential for the reality condition. The limitation of the momentum-space formula makes it urgent to search for a local characterization of real topological phases in systems with disorder and more generally in open-boundary systems inherently lacking translation and $\mathcal{I}_{ST}$ symmetries, such as quasicrystals [70–75] and amorphous systems [76–81].

In this Letter, we develop a real-space formalism for Euler class topology in 2D systems. In an analogy to the Chern class, we introduce a local Euler marker $e(\mathbf{r})$ to directly map the Euler topology in real space for both crystals and noncrystalline systems. The macroscopic average of $e(\mathbf{r})$ coincides with the Euler number regardless of periodic or open boundary conditions. We validate our real-space formalism by verifying topological Euler and trivial phases in clean systems, yielding consistent results with $\mathbf{k}$-space approaches. Additionally, we apply our method to a particular $\mathcal{PT}$-symmetric disordered system, successfully diagnosing the disorder-induced topological phase transition. Furthermore, our real-space formalism proves powerful in characterizing fragile topological phases in quasicrystals and even in amorphous systems lacking any spatial symmetries.

## 2 Characteristic class in k- versus r-space

The Euler class is a characteristic class of oriented real vector bundles. It can be constructed using an orthonormal basis $\{|u_n(\mathbf{k})\rangle\}$, where $|u_n(\mathbf{k})\rangle$ represents the cell-periodic part of the $n$-th occupied Bloch state $\langle \mathbf{r}|\psi_n(\mathbf{k})\rangle = e^{i\mathbf{k}\cdot\mathbf{r}}\langle \mathbf{r}|u_n(\mathbf{k})\rangle$. Utilizing this basis, we obtain the curvature matrix $\mathcal{F}$ with its entries given by:

$$\mathcal{F}_{mn}(\mathbf{k}) = \langle \partial_{[k_x} u_m(\mathbf{k})|\partial_{k_y]} u_n(\mathbf{k})\rangle \mathrm{d}k_x \wedge \mathrm{d}k_y \,, \tag{1}$$

where $[\cdots,\cdots]$ denotes the commutator applied to the index $k_x$ and $k_y$. When there are two occupied bands, the Euler class can be expressed as a differential 2-from in $\mathbf{k}$ space,

$$\begin{aligned} e(\mathcal{F}) &= \frac{1}{2\pi}\mathrm{Pf}(\mathcal{F}) \\ &= \frac{1}{2\pi}\langle \partial_{[k_x} u_1(\mathbf{k})|\partial_{k_y]} u_2(\mathbf{k})\rangle \mathrm{d}k_x \wedge \mathrm{d}k_y \,, \end{aligned} \tag{2}$$

where Pf denotes the Pfaffian acting on the matrix $\mathcal{F}$. The Euler number $e$ is an integer topological invariant for two real bands, which can be expressed as a simple $\mathbf{k}$-space integral [82],

$$e = \frac{1}{2\pi}\int_{BZ} \langle \partial_{[k_x} u_1(\mathbf{k})|\partial_{k_y]} u_2(\mathbf{k})\rangle \mathrm{d}k_x \mathrm{d}k_y \,. \tag{3}$$

To derive the expression of the Euler number in $r$-space, we start by replacing the occupied states in the above expression with a projection operator $\hat{P}(\mathbf{k}) = \sum_{\text{occ}} |u_n(\mathbf{k})\rangle\langle u_n(\mathbf{k})|$ in the occupied subspace [11]. After some algebra (see appendix B), we obtain the $\mathbf{k}$-space formula of Euler number $e$ represented by $\hat{P}(\mathbf{k})$,

$$e = \frac{1}{2\pi} \int_{BZ} \mathrm{d}^2\mathbf{k} \, \mathrm{Pf}_{\text{occ}}(\hat{P}(\mathbf{k})[\partial_{k_x}\hat{P}(\mathbf{k}), \partial_{k_y}\hat{P}(\mathbf{k})]), \tag{4}$$

where $\mathrm{Pf}_{\text{occ}}$ denotes the Pfaffian taken over the occupied subspace.

To generalize a formula of topological system defined in $\mathbf{k}$ space to its real-space form applicable to disordered system, a standard mathematical framework is the non-commutative geometry [83], which provides the duality (see the equivalence at least for translational invariant systems in appendix C),

$$\int_{BZ} \frac{\mathrm{d}^2\mathbf{k}}{(2\pi)^2/A} \to \mathrm{Tr},$$
$$\partial_{k_x}\hat{P}(\mathbf{k}) \to \frac{L_x}{2\pi}(\hat{U}\hat{P}\hat{U}^\dagger - \hat{P}), \tag{5}$$
$$\partial_{k_y}\hat{P}(\mathbf{k}) \to \frac{L_y}{2\pi}(\hat{V}\hat{P}\hat{V}^\dagger - \hat{P}),$$

where $A = L_x L_y$ is the area of the system, $\hat{U} = \exp(2\pi \mathrm{i}\hat{X}/L_x)$ and $\hat{V} = \exp(2\pi \mathrm{i}\hat{Y}/L_y)$ are the unitary position operator, Tr is the trace over the coordinate space, and $\hat{P}$ is the $r$-space projection operator. Note that the order of $\hat{P}$ is determined by both the site coordinates $r_i = (x_i, y_i)$, dependent on the lattice size, and the internal index $n$, matching the order of $\hat{P}(\mathbf{k})$. Therefore, we can divide the space on which $\hat{P}$ operates into two subspaces, $S(\hat{P}) = l^2(\mathbb{T}^2) \otimes \mathbb{R}^N$. Here, $l^2(\mathbb{T}^2)$ is the coordinate space, where $\mathbb{T}^2$ denotes the two-torus, a rectangle with edge length $L_x$ and $L_y$ with periodic boundary conditions (PBC) [14]. And $\mathbb{R}^N$ is internal space with the internal degrees of freedom $N$ which are those degrees of freedom except for the coordinate $\mathbf{k}$ or $\{r_i\}$. Consequently, we arrive at the $r$-space expression for the Euler number:

$$e = \frac{1}{2\pi} \mathrm{Tr} \mathrm{Pf}_{\text{occ}}(\hat{P}[\hat{U}\hat{P}\hat{U}^\dagger, \hat{V}\hat{P}\hat{V}^\dagger]), \tag{6}$$

where $\mathrm{Pf}_{\text{occ}}$ denotes the Pfaffian taken over the occupied submatrix in the internal space (see appendix D for more details). Formally, Eq. (6) share a similar expression to the real-space Chern number except for the substituting from Tr to $\mathrm{Pf}_{\text{occ}}$. Thus analogous to prior work on the local Chern marker [11], we propose defining the local Euler marker $e(\mathbf{r})$ as the expression in Eq. (6) before taking the trace, i.e.,

$$e(\mathbf{r}) = \frac{1}{2\pi} \mathrm{Pf}_{\text{occ}}(\langle\mathbf{r}|\hat{P}[\hat{U}\hat{P}\hat{U}^\dagger, \hat{V}\hat{P}\hat{V}^\dagger]|\mathbf{r}\rangle), \tag{7}$$

where $|\mathbf{r}\rangle$ denotes the basis to construct the external space indexed by the Wannier cell $\mathbf{r}$. The $r$-space Euler number (6) and local Euler marker (7) apply well to both crystalline and noncrystalline systems. They not only provide an intuitive local perspective of global topology but also serve as a valuable tool for distinguishing topological phases in aperiodic systems without translational symmetry.

## 3  Remarks on r-space Euler number

Before proceeding, we have a few remarks. First, the analysis we've conducted thus far can be directly applied to the Chern class, and the resultant $r$-space expression is nothing but

the Bott index, $\text{Bott}(\hat{U}, \hat{V}) = (1/2\pi)\text{ImTr}\log(\hat{U}\hat{V}\hat{U}^{-1}\hat{V}^{-1})$ with $\hat{U} = \hat{P}\exp(2\pi i\hat{X}/L_x)\hat{P}$ and $\hat{V} = \hat{P}\exp(2\pi i\hat{Y}/L_y)\hat{P}$, which offers an equivalent topological classification to the Chern number [14,15]. However, there are significant differences between the $r$-space formulation of the Euler and Chern number. The $r$-space Chern number only requires a simple trace performed consistently in both coordinate and internal space. In contrast, for the $r$-space Euler number, it becomes essential to distinguish between the coordinate and internal space, which requires trace and Pfaffian operations, respectively.

Secondly, to decompose the coordinate and internal spaces for extracting the occupied submatrix needed for Pfaffian calculation, we apply a unitary transformation to the eigenstates which makes $\hat{P}$ block-diagonal. This unitary transformation corresponds to constructing a set of composite Wannier functions, which can be determined by an explicit algorithm of localization functional minimization proposed by Marzari and Vanderbilt [84,85] (see appendix H.5). Importantly, while a nontrivial topological invariant may pose a topological obstruction for constructing Wannier representations composed of exponentially localized states in line with lattice symmetries [49, 51, 86, 87], it does not hinder the search for composite Wannier functions with optimal power-law decay [88–93].

Thirdly, the distinct treatments of Chern and Euler numbers in real space also lead to different behaviors in finite samples under open boundary conditions (OBC). It's well-known that the summation of the local Chern marker over an entire open system must equal zero, regardless of whether the system is a Chern insulator or not. This is because the local Chern marker in the bulk is always offset by the significant deviation at the boundary [11, 14]. In contrast, the local Euler marker near the open boundary fades away and thus doesn't suffer from the counteraction under OBC, making the choice of boundary condition irrelevant for the $r$-space Euler number (see appendix E).

## 4 Tight-binding model

To numerically validate the $r$-space formula of Euler number, we consider a general $\mathcal{PT}$-symmetric tight-binding model with the basis $(ip_x, ip_y, d_{xy}, d_{x^2-y^2})$ per site. The Hamiltonian is given by

$$H = \sum_{i\mu} \epsilon_\mu c_{i\mu}^\dagger c_{i\mu} + \sum_{\langle ij \rangle} \sum_{\mu\nu} t_{\mu\nu}(\boldsymbol{r}_{ij}) c_{i\mu}^\dagger c_{j\nu}, \tag{8}$$

where $c_{i\mu}^\dagger(c_{i\mu})$ is electron creation (annihilation) operator on the $\mu$ orbital at the $i$-th site. $\epsilon_\mu$ is the on-site energy and $t_{\mu\nu}(\boldsymbol{r}_{ij})$ is the Slater-Koster parameterized hopping integral [94,95] and has an inverse-square decay with the distance (i.e., $|\boldsymbol{r}_{ij}|^{-2}$) [96] (See details in appendix H.1). It has been proven that a $\mathcal{PT}$-symmetric Hamiltonian can become real-valued through the Takagi decomposition [49, 97]. Here we intentionally chose the $p$ orbitals to be imaginary, which results in $\mathcal{PT} = \hat{K}$ with the complex conjugation operator $\hat{K}$. The invariance of the Hamiltonian under $\mathcal{PT}$ imposes the reality condition on $H$. It was previously known that a fragile topological state with a nontrivial Euler number $e = 1$ can be achieved by considering a double band inversion between $p_{x,y}$ and $d_{x^2-y^2,xy}$ orbitals [98]. Here we verify the validity of the $r$-space Euler number in both crystalline and noncrystalline systems based on this model. We also validate our expression using other models with different Euler numbers, which are detailed in the appendix I.8.

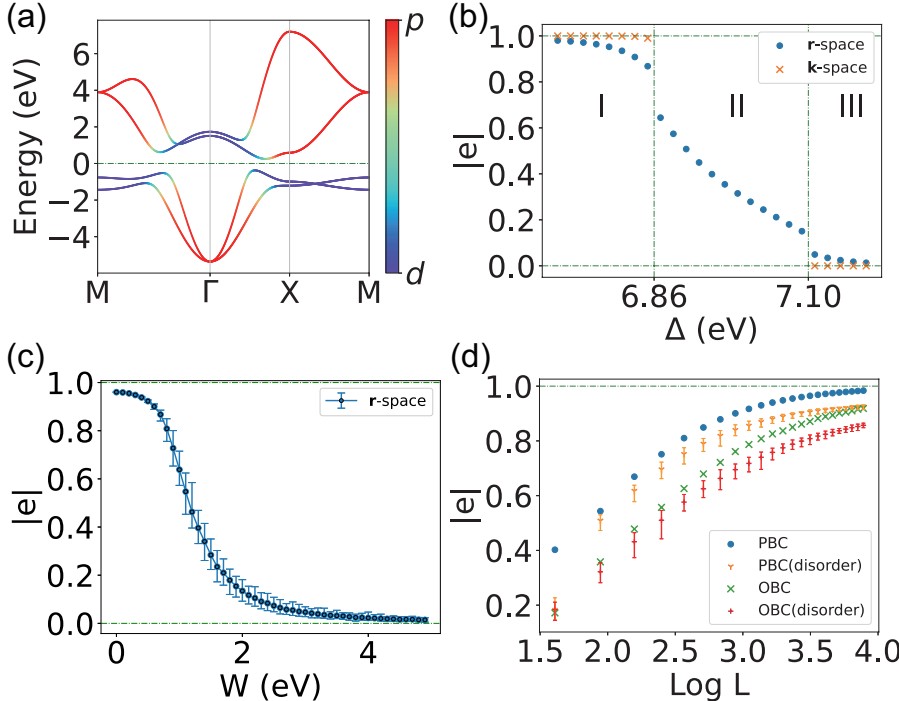

Figure 1: (a) Orbital-resolved band structures of the square lattice with a double band inversion between $p_{x,y}$ and $d_{x^2-y^2,xy}$ orbitals. The parameters used are $\epsilon_{p_x,p_y} = 1.58$, $\epsilon_{d_{x^2-y^2,xy}} = -0.42$, $V_{pp\sigma}=-0.865$, $V_{pp\pi}=-0.144$, $V_{pd\sigma}=0.173$, $V_{pd\pi}=0.135$, $V_{dd\sigma}=0.144$, $V_{dd\pi}=0.124$, $V_{dd\delta}=0.259$ eV. (b) The variation of the Euler number as the on-site energy difference $\Delta = \epsilon_p - \epsilon_d$ changes. Other parameters remain unchanged and the lattice size is $L = 201$. (c) The $\boldsymbol{r}$-space Euler number as a function of the disorder strength $W$ in $31 \times 31$ square lattices with periodic boundary condition (PBC). (d) The lattice size $L$ dependence of the $\boldsymbol{r}$-space Euler number calculated without and with on-site energy disorder ($W = 1.0$ eV) using PBC and open boundary condition (OBC). For each $L$ and $W$, the configuration average is performed over 100 realizations.

## 5 Diagnosis of topological phase transitions

With the well-defined $\boldsymbol{r}$-space Euler number, we first diagnose topological phase transitions in a square lattice based on the model in Eq. (8). As shown in Fig. 1(a), the orbital-resolved band structure displays signs of a double band inversion between $p_{x,y}$ and $d_{x^2-y^2,xy}$ orbitals around the $\Gamma$ point, implying their nontrivial electronic topology. We compute the Euler number in both $\boldsymbol{k}$-space and $\boldsymbol{r}$-space, consistently yielding a value of $e = 1$, thus confirming the nontrivial Euler topology. We further examine the evolution of the Euler number in both $\boldsymbol{k}$- and $\boldsymbol{r}$-spaces with increasing the on-site energy difference $\Delta = \epsilon_p - \epsilon_d$. In Fig. 1(b), the system undergoes a topological phase transition from a topological Euler insulator with $e = 1$ (region I) to an intermediate gapless state (II) and eventually transitions into a trivial insulator with $e = 0$ (III). The calculated $\boldsymbol{r}$-space Euler number matches with the $\boldsymbol{k}$-space one, except for the intermediate gapless phase (region II) where the Euler number is ill-defined. This transition can be understood by tracing the evolution of band inversion (see appendix I.1): Starting from a double inverted band order, the nontrivial energy gap gradually decreases to zero with increasing $\Delta$, then remains closed over a finite $\Delta$ range, and eventually reopens with a trivial normal band order.

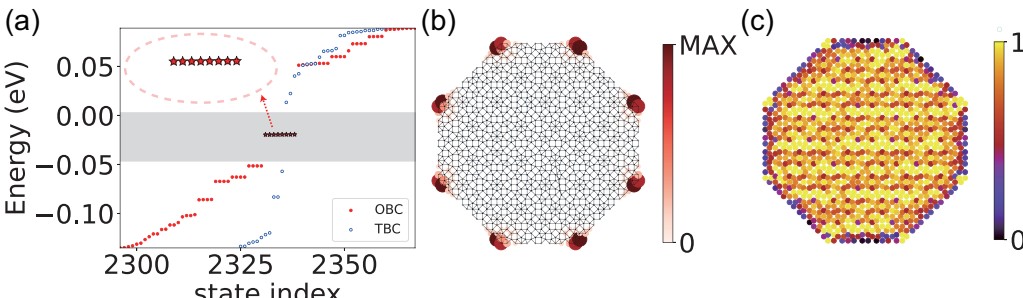

Figure 2: Fragile topological state characterized by $e = 1$ in the Ammann-Beenker-tiling quasicrystal based on the model in Eq. (8). Parameters are $\epsilon_{p_x, p_y} = 1.58$, $\epsilon_{d_{x^2-y^2}, xy} = $ -0.42, $V_{pp\sigma} = -1.783, V_{pp\pi} = -0.299, V_{pd\sigma} = 0.359, V_{pd\pi} = 0.280$, $V_{dd\sigma} = 0.299, V_{dd\pi} = 0.257, V_{dd\delta} = 0.537$ eV. (a) Energy spectrum of the quasicrystal containing 1168 sites with OBC or twisted boundary condition (TBC). Insert shows 8 corner states (highlighted by red stars) in the bulk gap. (b) Spatial distribution of the in-gap corner states [red stars in (a)]. (c) The distribution of local Euler markers $e(\mathbf{r})$ in the quasicrystal with OBC.

Next, we demonstrate the applicability of the $\mathbf{r}$-space Euler number for aperiodic systems by introducing the disorder term in the on-site energies of the aforementioned model. We specifically consider disorder term that preserves $\mathcal{PT}$ symmetry, which is represented by $V_{\text{dis}} = \sum_{i \in \tau_{1/2}} \lambda_i (c_i^\dagger c_i + c_{\mathcal{P}i}^\dagger c_{\mathcal{P}i})$ with the random variables $\{\lambda_i\}$ distributed uniformly within the interval $[-W, W]$ on half of the sites ($\tau_{1/2}$) in the sample, where $W$ is the disorder strength. The annihilation operators $c_i$ and $c_{\mathcal{P}i}$ act on the site at $\mathbf{r}_i$ and its inversion partner $\mathcal{P}\mathbf{r}_i$, respectively. The averaged $\mathbf{r}$-space Euler number as a function of $W$ is shown in Fig. 1(c). For moderate disorder, the $\mathbf{r}$-space Euler number $e$ remains around 1, indicating the system remains topologically nontrivial. Remarkably, as disorder strength $W$ increases, $e$ gradually decreases to 0, diagnosing a topological phase transition (see appendix I.5). Our results confirm the disorder-induced topological phase transition classified by the topological Euler class [99,100], and validate the $\mathbf{r}$-space formalism of Euler number in disordered systems.

We further check the effect of lattice size and different boundary conditions on the $\mathbf{r}$-space Euler number, as shown in Fig. 1(d). All calculated $\mathbf{r}$-space Euler numbers converge to the limit of 1 with different rates by increasing the lattice size, demonstrating the faithful formalism of the Euler number. Importantly, the OBC results exhibit a deviation from PBC due to the presence of open boundaries, but this difference can be diminished by increasing lattice size (see appendix I.4).

This suggests that the $\mathbf{r}$-space formula remains reliable regardless of the boundary conditions, which is notably different from the Chern number.

It is also noted that the disordered case converges much slower than the pristine PBC case. Because the disordered system is close to the critical point, the energy gap reduces significantly and the correlation length increases, which demands larger lattice sizes for accurate calculations of the real-space Euler number $^0$. Our results show that the $\mathbf{r}$-space Euler number equals the exact one within a correction of order $\mathcal{O}(1/(L\Delta E))$ for systems with lattice size $L$ and energy gap $\Delta E$, which resembles the case of Bott index and Chern number [14].

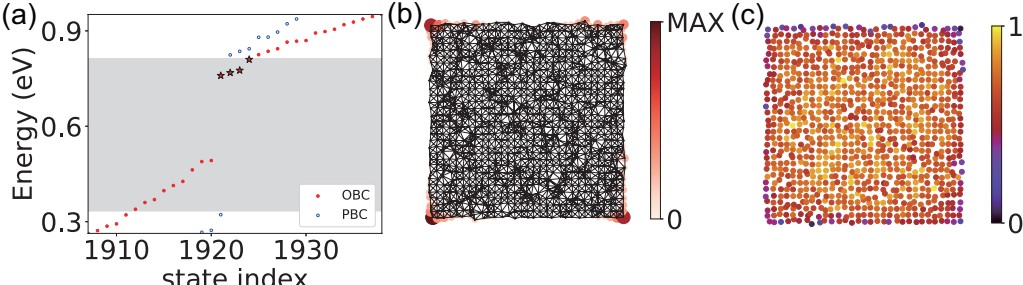

Figure 3: Fragile topological state characterized by $e = 1$ in the amorphous square lattice based on the model in Eq. (8). Each atom is assigned with a random displacement following the Gaussian distribution with standard deviation $\sigma = 0.2$. Parameters are $L$=31, $\epsilon_{p_x,p_y} = 1.58$, $\epsilon_{d_{x^2-y^2,xy}} = $-0.42, $V_{pp\sigma}$=-0.565, $V_{pp\pi}$=-0.044, $V_{pd\sigma}$=0.773, $V_{pd\pi}$=0.335, $V_{dd\sigma}$=0.444, $V_{dd\pi}$=0.224, $V_{dd\delta}$=0.659 eV. (a) Energy spectrum of the amorphous square lattice with PBC and OBC. Four corner states in the gap are highlighted by red stars. (b) Spatial distribution of the corner states [red stars in (a)]. (c) The distribution of $e(\mathbf{r})$ for the amorphous system with OBC.

# 6  Fragile topology in quasicrystals and amorphous lattices

As an application of our proposed $\mathbf{r}$-space formula, we explore the Euler topology in quasicrystals and amorphous lattices. Specifically, we consider the 2D Ammann-Beenker-tiling quasicrystal, which possesses 8-fold rotational symmetry but lacks transitional symmetry. In the finite octagonal quasicrystal sample with open boundary conditions (OBC), 8 degenerate states emerge within the bulk gap region (grey area), as shown in Fig. 2(a). The bulk gap estimation utilizes a twisted boundary condition (TBC) to preserve octagonal symmetry and eliminate boundary effects (see appendix H.4). We plot the spatial distribution of these in-gap states [see Fig. 2(b)], and find that they are well localized at 8 corners of the octagonal quasicrystal, implying its feature of higher-order topology. We also examine the local Euler marker distribution in the finite quasicrystal sample, as depicted in Fig. 2(c). The plot confirms that the local Euler markers $e(\mathbf{r})$ closely match the expected value of 1 within the bulk but deviate at the edges. As expected, the average of $e(\mathbf{r})$ over the entire finite sample does not vanish but yields $e \approx 1$, verifying the nontrivial Euler topology of the quasicrystal.

We further study a finite amorphous lattice constructed by assigning random site displacements away from their equilibrium position in an initial square lattice. Consequently, all spatial symmetries are broken, including $\mathcal{P}$ or $\mathcal{C}_{2z}$ demanded by $\mathcal{I}_{ST}$ symmetry for real Bloch states in periodic crystals. Nevertheless, for the spinless model (8) in any amorphous lattice with OBC, it is always possible to choose a real gauge so that both the Hamiltonian and eigenstates can be taken real (see appendix G). This implies that the $\mathbf{r}$-space Euler number is still applicable to identify its Euler topology. As shown in Fig. 3(a), the energy spectrum of the finite amorphous lattice with OBC exhibits 4 corner states at the Fermi level in the bulk gap estimated using artificial PBC (grey area). The spatial distribution of these states supports that they are indeed localized at 4 corners of the finite sample [see Fig. 3(b)]. As shown in Fig. 3(c), local Euler markers $e(\mathbf{r})$ are dominated in the internal area but tend to vanish at the boundary of the finite amorphous sample. The sum of $e(\mathbf{r})$ over the entirety of the finite sample yields a nonzero Euler number which is expected to converge to the quantized value of 1 with increasing lattice size.

## 7 Conclusion and discussion

We have proposed an explicit real-space formula for the Euler number to identify the fragile topological phases in both crystalline and noncrystalline systems whose wave functions are real. Specifically, the local Euler marker $e(\mathbf{r})$ whose macroscopic average coincides with the Euler number $e$, is introduced to characterize the topological order in real space. Notably, this applies equally well to periodic and open boundary conditions. We have validated our expression by diagnosing the topological phase transition in crystals and disordered systems with $\mathcal{PT}$ symmetry. Furthermore, we have also uncovered the topological Euler phases in quasicrystals and amorphous lattices without any spatial symmetry. Our work greatly extends the concept of real-space topological markers to topological states in real Hilbert space and would hopefully inspire future exploration in more topological characteristic classes in real space.

Despite the progress made, several critical issues remain open for further investigation. Given the multi-gap nature of the Euler number, it is essential to develop real-space Wannier functions that can effectively disentangle the internal space from the full system during numerical calculations. Rigorously defining the Pfaffian marker in real space without reference to translationally invariant cases remains a challenging task. While the main text presents several examples, it does not yet explore a purely amorphous case that operates independently of any translationally invariant lattices. Additionally, a comprehensive mathematical framework has yet to be fully developed. We hope this work inspires future research efforts aimed at applying tools from non-commutative geometry to address the intricate challenges associated with the Euler number.

## Acknowledgments

We thank Guo Chuan Thiang for the valuable discussions. We also would like to express our sincere gratitude to the referees for their careful review of our manuscript and their constructive comments. Their insightful feedback and suggestions significantly contributed to the improvement of the overall quality of this work. We appreciate their time and effort in providing such thoughtful evaluations.

**Author contributions**   D.L. and C.W. contributed equally to this work.

**Funding information**   This work is supported by the National Key R&D Program of China (Grant No. 2021YFA1401600) and the National Natural Science Foundation of China (Grant No. 12074006 and 12474056). The computational resources were supported by the high-performance computing platform of Peking University.

## A   Orientability of our models

In this section, we examine the orientability of our models. The Euler class $\mathfrak{e}(\mathcal{F})$ is defined as

$$\mathfrak{e}(\mathcal{F}) = \frac{1}{2\pi}\mathrm{Pf}(\mathcal{F}), \tag{A.1}$$

where Pf denotes the Pfaffian acting on the curvature matrix $\mathcal{F}$. Under the basis transformation $O$, the Euler class acquires an additional factor $\det(O)$, as shown below:

$$
\begin{aligned}
\mathfrak{e}(\mathcal{F}) &\to \mathfrak{e}(O^{-1}\mathcal{F}O) \\
&= \frac{1}{2\pi}\text{Pf}(O^{-1}\mathcal{F}O) \\
&= \frac{1}{2\pi}\text{Pf}(O^{T}\mathcal{F}O),
\end{aligned}
\tag{A.2}
$$

where the last equality originates from the orthonormality property of the real wave functions, which means $O^{-1} = O^{T}$. It's worth noting that for a $2n \times 2n$ skew-symmetric matrix $A$ and an arbitrary $2n \times 2n$ matrix $B$, the Pfaffian satisfies the identity $\text{Pf}(B^{T}AB) = \text{Pf}(A)\det(B)$. Therefore, since the curvature matrix $\mathcal{F}$ is skew-symmetric, we can simplify the expression further as:

$$
\begin{aligned}
\mathfrak{e}(\mathcal{F}) &\to \frac{1}{2\pi}\text{Pf}(\mathcal{F})\det(O) \\
&= \mathfrak{e}(\mathcal{F})\det(O).
\end{aligned}
\tag{A.3}
$$

For the Euler class $\mathfrak{e}(\mathcal{F})$ to be a characteristic class, it must remain invariant under any basis transformation. Therefore, a certain transformation matrix $O$ with $\det(O) = 1$ is essential. Since $O$ is the transformation matrix between orthonormal basis, it naturally satisfies the condition $|\det(O)|=1$. Thus, system orientability is necessary to prevent $\det(O) = -1$ and ensure the invariance of the Euler class.

In fact, the orientability of the Brillouin zone is determined by the first Stiefel-Whitney class $w_1$, which is the total Berry phase of the occupied states over the Brillouin zone [45]. Because the Chern number of a time-reversal symmetric system is always trivial, a complex smooth gauge can be found in this system. Given a Berry connection $A$ that satisfies $\mathcal{F} = dA$ in this gauge, we have

$$
w_1|_{C} = \frac{1}{\pi}\oint_{C} d\boldsymbol{k} \cdot \text{Tr}A(\boldsymbol{k}).
\tag{A.4}
$$

Therefore, our models are easily confirmed to be orientable with a trivial $w_1 = 0$, allowing us to proceed with our discussion on the Euler class and the second Stiefel-Whitney class.

## B  Derivation of Eq. (4) in the main text

In this section, we derive Eq. (4) in the main text, beginning with the relation between the Chern and Euler class in a two-dimensional system. Specifically, there is a correspondence between the first Chern class $c_1$ and the Euler class $\mathfrak{e}$:

$$
c_1(\mathcal{F}_{\mathbb{C}}) = \mathfrak{e}(\mathcal{F}),
\tag{B.1}
$$

where $\mathcal{F}_{\mathbb{C}}$ is the curvature over a complex number field, isomorphic to $\mathcal{F}$ over a real number field through an isomorphism $\mathbb{C} \cong \mathbb{R} \oplus \mathbb{R}$. In particular, for a system with two occupied bands ($N_{\text{occ}} = 2$), we can construct a complex Bloch state

$$
|u\rangle = \frac{1}{\sqrt{2}}(|u_1\rangle + i|u_2\rangle),
\tag{B.2}
$$

where $|u_n\rangle$ ($n = 1, 2$) represents the cell-periodic part of the $n$-th occupied Bloch state $|\psi_n(\boldsymbol{k})\rangle$. Note that for brevity, we omit the explicit dependence of $\boldsymbol{k}$ in this section for $|u_n(\boldsymbol{k})\rangle$, $|u(\boldsymbol{k})\rangle$,

and the projection operator $\tilde{P}(\boldsymbol{k})$. Based on the complex Bloch states, the first Chern class is given by

$$c_1(\mathcal{F}_{\mathbb{C}}) = \frac{1}{2\pi \mathrm{i}}\mathcal{F}_{\mathbb{C}} = \frac{1}{2\pi \mathrm{i}}\langle \partial_{[k_x}u|\partial_{k_y]}u\rangle \mathrm{d}k_x \wedge \mathrm{d}k_y\,. \tag{B.3}$$

This allows us to derive the expression of the Euler class from the first Chern class.

To begin with, we can express the first Chern number as a $\boldsymbol{k}$-space integral:

$$c_1 = \frac{1}{2\pi \mathrm{i}}\int_{BZ} \mathrm{d}^2\boldsymbol{k}\,\mathrm{Tr}(\tilde{P}\partial_{[k_x}\tilde{P}\partial_{k_y]}\tilde{P})\,, \tag{B.4}$$

where the integral is over the Brillouin zone (BZ) and $\tilde{P} = |u\rangle\langle u|$ is the projection operator, with its real and imaginary parts given by:

$$\mathrm{Re}\tilde{P} = \frac{1}{2}(|u_1\rangle\langle u_1| + |u_2\rangle\langle u_2|)\,, \tag{B.5}$$

and

$$\mathrm{Im}\tilde{P} = \frac{1}{2}(|u_2\rangle\langle u_1| - |u_1\rangle\langle u_2|)\,. \tag{B.6}$$

Using Eq. (B.2), we can rewrite Eq. (B.4) as

$$c_1 = \frac{1}{2\pi \mathrm{i}}\int_{BZ} \mathrm{d}^2\boldsymbol{k}\,\langle u|[\partial_{k_x}\tilde{P}, \partial_{k_y}\tilde{P}]|u\rangle\,, \tag{B.7}$$

and then the Euler number is given by

$$e = \frac{1}{4\pi \mathrm{i}}\int_{BZ} \mathrm{d}^2\boldsymbol{k}\,\langle u_1|[\partial_{k_x}\tilde{P}, \partial_{k_y}\tilde{P}]|u_1\rangle + \frac{1}{4\pi \mathrm{i}}\int_{BZ} \mathrm{d}^2\boldsymbol{k}\,\langle u_2|[\partial_{k_x}\tilde{P}, \partial_{k_y}\tilde{P}]|u_2\rangle$$
$$+ \frac{1}{4\pi}\int_{BZ} \mathrm{d}^2\boldsymbol{k}\,\langle u_1|[\partial_{k_x}\tilde{P}, \partial_{k_y}\tilde{P}]|u_2\rangle - \frac{1}{4\pi}\int_{BZ} \mathrm{d}^2\boldsymbol{k}\,\langle u_2|[\partial_{k_x}\tilde{P}, \partial_{k_y}\tilde{P}]|u_1\rangle\,. \tag{B.8}$$

To keep the Euler number $e$ real, we can simplify the operators $[\partial_{k_x}\tilde{P}, \partial_{k_y}\tilde{P}]$ in Eq. (B.8) to

$$\mathrm{i}[\partial_{k_x}\mathrm{Re}\tilde{P}, \partial_{k_y}\mathrm{Im}\tilde{P}] + \mathrm{i}[\partial_{k_x}\mathrm{Im}\tilde{P}, \partial_{k_y}\mathrm{Re}\tilde{P}]\,, \tag{B.9}$$

for the first two terms and

$$[\partial_{k_x}\mathrm{Re}\tilde{P}, \partial_{k_y}\mathrm{Re}\tilde{P}] - [\partial_{k_x}\mathrm{Im}\tilde{P}, \partial_{k_y}\mathrm{Im}\tilde{P}]\,, \tag{B.10}$$

for the other terms. Since $\{|u_n\rangle\}$ are orthonormal, we have the following identities:

$$\langle u_n|u_m\rangle = \delta_{n,m}\,, \tag{B.11}$$

and

$$\langle u_n|\partial_{k_i}u_n\rangle = \frac{1}{2}\partial_{k_i}(\langle u_n|u_n\rangle) = 0\,. \tag{B.12}$$

Therefore, we have

$$\begin{cases} \partial_{k_i}\mathrm{Re}\tilde{P}|u_1\rangle = \dfrac{1}{2}(|\partial_{k_i}u_1\rangle + |u_2\rangle\langle u_1|\partial_{k_i}u_2\rangle)\,, \\[2mm] \partial_{k_i}\mathrm{Re}\tilde{P}|u_2\rangle = \dfrac{1}{2}(|\partial_{k_i}u_2\rangle - |u_1\rangle\langle u_1|\partial_{k_i}u_2\rangle)\,, \\[2mm] \partial_{k_i}\mathrm{Im}\tilde{P}|u_1\rangle = \dfrac{1}{2}(|\partial_{k_i}u_2\rangle - |u_1\rangle\langle u_1|\partial_{k_i}u_2\rangle) \;\; = \partial_{k_i}\mathrm{Re}\tilde{P}|u_2\rangle\,, \\[2mm] \partial_{k_i}\mathrm{Im}\tilde{P}|u_2\rangle = -\dfrac{1}{2}(|\partial_{k_i}u_1\rangle + |u_2\rangle\langle u_1|\partial_{k_i}u_2\rangle) = -\partial_{k_i}\mathrm{Re}\tilde{P}|u_1\rangle\,, \end{cases} \tag{B.13}$$

with $k_i$ denoting $k_x$ or $k_y$. Since $\mathrm{Re}\tilde{P}$ is a Hermitian operator and $\mathrm{Im}\tilde{P}$ is an anti-Hermitian operator, we have:

$$\langle u_n|\partial_{k_i}\mathrm{Re}\tilde{P} = (\partial_{k_i}\mathrm{Re}\tilde{P}|u_n\rangle)^\dagger\,, \tag{B.14}$$

and

$$-\langle u_n|\partial_{k_i}\mathrm{Im}\tilde{P} = (\partial_{k_i}\mathrm{Im}\tilde{P}|u_n\rangle)^\dagger\,, \tag{B.15}$$

where the additional minus sign in Eq. (B.15) can be canceled by the minus sign in the commutators in Eq. (B.9) and Eq. (B.10).

Therefore, the first term in Eq. (B.8) is

$$\frac{1}{4\pi\mathrm{i}}\int_{BZ}\mathrm{d}^2\boldsymbol{k}\,\langle u_1|[\partial_{k_x}\tilde{P},\partial_{k_y}\tilde{P}]|u_1\rangle = \frac{1}{4\pi}\int_{BZ}\mathrm{d}^2\boldsymbol{k}\,\langle u_1|([\partial_{k_x}\mathrm{Re}\tilde{P},\partial_{k_y}\mathrm{Im}\tilde{P}]+[\partial_{k_x}\mathrm{Im}\tilde{P},\partial_{k_y}\mathrm{Re}\tilde{P}])|u_1\rangle$$

$$= \frac{1}{2\pi}\int_{BZ}\mathrm{d}^2\boldsymbol{k}(\langle u_1|\partial_{k_x}\mathrm{Re}\tilde{P}\partial_{k_y}\mathrm{Im}\tilde{P}|u_1\rangle - \langle u_1|\partial_{k_y}\mathrm{Re}\tilde{P}\partial_{k_x}\mathrm{Im}\tilde{P}|u_1\rangle)$$

$$= \frac{1}{2\pi}\int_{BZ}\mathrm{d}^2\boldsymbol{k}(\langle u_1|\partial_{k_x}\mathrm{Re}\tilde{P}\partial_{k_y}\mathrm{Re}\tilde{P}|u_2\rangle - \langle u_1|\partial_{k_y}\mathrm{Re}\tilde{P}\partial_{k_x}\mathrm{Re}\tilde{P}|u_2\rangle)$$

$$= \frac{1}{2\pi}\int_{BZ}\mathrm{d}^2\boldsymbol{k}\,\langle u_1|[\partial_{k_x}\mathrm{Re}\tilde{P},\partial_{k_y}\mathrm{Re}\tilde{P}]|u_2\rangle\,. \tag{B.16}$$

The analysis of the second term in Eq. (B.8) is similar, with the only difference being an additional minus sign from Eq. (B.15) as

$$\frac{1}{4\pi\mathrm{i}}\int_{BZ}\mathrm{d}^2\boldsymbol{k}\,\langle u_2|[\partial_{k_x}\tilde{P},\partial_{k_y}\tilde{P}]|u_2\rangle = -\frac{1}{2\pi}\int_{BZ}\mathrm{d}^2\boldsymbol{k}\,\langle u_2|\partial_{[k_x}\mathrm{Re}\tilde{P}\partial_{k_y]}\mathrm{Re}\tilde{P}|u_1\rangle$$

$$= \frac{1}{2\pi}\int_{BZ}\mathrm{d}^2\boldsymbol{k}\,\langle u_1|[\partial_{k_x}\mathrm{Re}\tilde{P},\partial_{k_y}\mathrm{Re}\tilde{P}]|u_2\rangle\,, \tag{B.17}$$

where the last equality holds due to the Hermiticity of $\mathrm{Re}\tilde{P}$ and the reality of $|u_n\rangle$. Now, let's consider the third term in Eq. (B.8), which is

$$\frac{1}{4\pi}\int_{BZ}\mathrm{d}^2\boldsymbol{k}\,\langle u_1|[\partial_{k_x}\tilde{P},\partial_{k_y}\tilde{P}]|u_2\rangle$$

$$= \frac{1}{4\pi}\int_{BZ}\mathrm{d}^2\boldsymbol{k}(\langle u_1|[\partial_{k_x}\mathrm{Re}\tilde{P},\partial_{k_y}\mathrm{Re}\tilde{P}]|u_2\rangle - \langle u_1|[\partial_{k_x}\mathrm{Im}\tilde{P},\partial_{k_y}\mathrm{Im}\tilde{P}]|u_2\rangle)$$

$$= \frac{1}{4\pi}\int_{BZ}\mathrm{d}^2\boldsymbol{k}(\langle u_1|[\partial_{k_x}\mathrm{Re}\tilde{P},\partial_{k_y}\mathrm{Re}\tilde{P}]|u_2\rangle - \langle u_2|[\partial_{k_x}\mathrm{Re}\tilde{P},\partial_{k_y}\mathrm{Re}\tilde{P}]|u_1\rangle)$$

$$= \frac{1}{2\pi}\int_{BZ}\mathrm{d}^2\boldsymbol{k}\,\langle u_1|[\partial_{k_x}\mathrm{Re}\tilde{P},\partial_{k_y}\mathrm{Re}\tilde{P}]|u_2\rangle\,. \tag{B.18}$$

Likewise, the final term in Eq. (B.8) can be expressed as:

$$\frac{1}{2\pi}\int_{BZ}\mathrm{d}^2\boldsymbol{k}\,\langle u_1|[\partial_{k_x}\mathrm{Re}\tilde{P},\partial_{k_y}\mathrm{Re}\tilde{P}]|u_2\rangle\,, \tag{B.19}$$

due to the anti-symmetry of $|u_1\rangle$ and $|u_2\rangle$.

Therefore, Eq. (B.8) is now simplified to

$$e = \frac{2}{\pi}\int_{BZ}\mathrm{d}^2\boldsymbol{k}\,\langle u_1|[\partial_{k_x}\mathrm{Re}\tilde{P},\partial_{k_y}\mathrm{Re}\tilde{P}]|u_2\rangle\,. \tag{B.20}$$

The relevant operator in the above expression is $\text{Re}\tilde{P}$. By introducing the real projector

$$\hat{P} := \sum_{n}^{\text{occ}} |u_n\rangle\langle u_n|\,, \tag{B.21}$$

we obtain the following identities:

$$\hat{P} = 2\text{Re}\tilde{P}\,, \tag{B.22}$$

and

$$\hat{P}|u_n\rangle = |u_n\rangle\,. \tag{B.23}$$

Thus, the formula Eq. (B.20) of the Euler number can be further expressed as:

$$e = \frac{1}{2\pi}\int_{BZ}\text{d}^2\boldsymbol{k}\,\langle u_1|\hat{P}[\partial_{k_x}\hat{P}, \partial_{k_y}\hat{P}]|u_2\rangle\,. \tag{B.24}$$

Due to the symmetry of $k_{x,y}$ and $|u_{1,2}\rangle$, the final form of the Euler number in $\boldsymbol{k}$-space is

$$e = \frac{1}{2\pi}\int_{BZ}\text{d}^2\boldsymbol{k}\,\text{Pf}_{\text{occ}}(\hat{P}[\partial_{k_x}\hat{P}, \partial_{k_y}\hat{P}])\,, \tag{B.25}$$

which is nothing but the Eq. (4) in the main text. Here $\text{Pf}_{\text{occ}}$ denotes the Pfaffian taken over the occupied subspace. To be specific, in the eigenbasis, a general matrix $\mathbf{M}$ can be represented as a block matrix

$$\mathbf{M} = \begin{pmatrix} M_1 & M_2 \\ M_3 & M_{\text{occ}} \end{pmatrix}\,, \tag{B.26}$$

where $M_{\text{occ}}$ is the submatrix of $\mathbf{M}$ constructed by occupied eigenbasis. Therefore, $\text{Pf}_{\text{occ}}$, which is the Pfaffian taken over the occupied subspace, is defined as

$$\text{Pf}_{\text{occ}}(M) := \text{Pf}(M_{\text{occ}})\,. \tag{B.27}$$

## C  Derivation of Eq. (6) in the main text

In this section, we derive Eq. (6) in the main text, demonstrating its equivalence to Eq. (4) in the main text under translational invariance.

Before proceeding, we first introduce some basic basis for the operators used in the derivation. Firstly, we use a $\boldsymbol{k}$-mesh form instead of the continuous form of the system. In real space, the Hamiltonian $\hat{H}$ is constructed under a certain initial local basis $\{|\alpha\boldsymbol{r}\rangle\}$ with $|\alpha\boldsymbol{r}\rangle = |\boldsymbol{r}\rangle\otimes|\alpha\rangle$, i.e.,

$$\hat{H} = \sum_{\alpha'\boldsymbol{r}', \alpha''\boldsymbol{r}''} |\alpha'\boldsymbol{r}'\rangle\langle\alpha''\boldsymbol{r}''|H_{\alpha'\boldsymbol{r}', \alpha''\boldsymbol{r}''}\,, \tag{C.1}$$

where $\alpha$ and $\boldsymbol{r}$ denote the internal and coordinate index, respectively. In $\boldsymbol{k}$ space, it is convenient to use the Bloch basis $\{|\psi_n(\boldsymbol{k})\rangle\}$ satisfying $|\psi_n(\boldsymbol{k})\rangle = |\boldsymbol{k}\rangle\otimes|u_n(\boldsymbol{k})\rangle$ where $\{|\boldsymbol{k}\rangle\}$ is the plane wave basis with $\langle\boldsymbol{r}|\boldsymbol{k}\rangle = \frac{1}{\sqrt{A}}e^{-i\boldsymbol{k}\cdot\boldsymbol{r}}$ and $A = L_x L_y$ being the area of the system. We can thus construct the $\boldsymbol{k}$-space Hamiltonian $\hat{H}(\boldsymbol{k})$ as

$$\begin{aligned} \hat{H}(\boldsymbol{k}) &= \langle\boldsymbol{k}|\hat{H}|\boldsymbol{k}\rangle \\ &= \sum_{\alpha', \alpha''} |\alpha'(\boldsymbol{k})\rangle\langle\alpha''(\boldsymbol{k})|H_{\alpha', \alpha''}(\boldsymbol{k}) \\ &= \sum_{n} |u_n(\boldsymbol{k})\rangle\langle u_n(\boldsymbol{k})|E_n(\boldsymbol{k})\,. \end{aligned} \tag{C.2}$$

Here, the second equality is established due to the translational invariance of the Hamiltonian. Additionally, the cell-periodic Bloch basis $\{|u_n(\boldsymbol{k})\rangle\}$ is the eigenbasis of $\hat{H}(\boldsymbol{k})$.

Then, we can define the projection operator acting on different basis sets as [85]

$$
\begin{aligned}
\hat{P} &= \sum_{n\boldsymbol{k}}^{\text{occ}} |\psi_n(\boldsymbol{k})\rangle\langle\psi_n(\boldsymbol{k})| \\
&= \sum_{\boldsymbol{k}} |\boldsymbol{k}\rangle\langle\boldsymbol{k}| \sum_{n}^{\text{occ}} |u_n(\boldsymbol{k})\rangle\langle u_n(\boldsymbol{k})| \\
&= \sum_{\boldsymbol{k}} |\boldsymbol{k}\rangle\langle\boldsymbol{k}| \sum_{\alpha',\alpha''} |\alpha'(\boldsymbol{k})\rangle\langle\alpha''(\boldsymbol{k})|P_{\alpha',\alpha''}(\boldsymbol{k}) \\
&= \sum_{\boldsymbol{k},\alpha',\alpha''} |\alpha'\boldsymbol{k}\rangle\langle\alpha''\boldsymbol{k}|P_{\boldsymbol{k},\alpha',\alpha''} \\
&= \sum_{\alpha'\boldsymbol{r}',\alpha''\boldsymbol{r}''} |\alpha'\boldsymbol{r}'\rangle\langle\alpha''\boldsymbol{r}''|P_{\alpha'\boldsymbol{r}',\alpha''\boldsymbol{r}''} \,.
\end{aligned}
\tag{C.3}
$$

So the $\boldsymbol{k}$-space projector $\hat{P}(\boldsymbol{k}) = \sum_n^{\text{occ}} |u_n(\boldsymbol{k})\rangle\langle u_n(\boldsymbol{k})|$ can be explicitly represented as a matrix $P(\boldsymbol{k})$ under basis $\{|\alpha(\boldsymbol{k})\rangle\}$. For convenience, we can create a new projection matrix $P_{\boldsymbol{k}}$, which is a quasi-diagonal matrix with $P(\boldsymbol{k})$ as diagonal blocks. In fact, $P_{\boldsymbol{k}}$ represents $\hat{P}$ under basis set $\{|\alpha\boldsymbol{k}\rangle\}$ and is related to $P$ under basis set $\{|\alpha\boldsymbol{r}\rangle\}$ via a unitary basis transformation. Specifically, we can construct a transformation matrix $U_{\boldsymbol{k},\boldsymbol{r}}$ with the entries as $\langle\boldsymbol{r}|\boldsymbol{k}\rangle$ to denote this basis transformation. Notice that $U_{\boldsymbol{k},\boldsymbol{r}}$ is indeed a unitary matrix in the thermodynamic limit $A \to \infty$. Therefore, we can obtain the $\boldsymbol{r}$-space projection matrix $P$ under the local basis by transforming $P_{\boldsymbol{k}}$ using the the transformation:

$$
P_{\boldsymbol{k}} = U_{\boldsymbol{k},\boldsymbol{r}} P U_{\boldsymbol{k},\boldsymbol{r}}^{\dagger} \,.
\tag{C.4}
$$

Now we start to derive Eq. (6). Since the integral is now discretized as

$$
\frac{A}{(2\pi)^2}\int_{BZ} \mathrm{d}^2\boldsymbol{k} \to \sum_{\boldsymbol{k}},
\tag{C.5}
$$

we can define its equivalent operation $\text{Tr}_{\boldsymbol{k}}$ acting on the block index $\boldsymbol{k}$ of $P_{\boldsymbol{k}}$. Therefore, the $\boldsymbol{k}$-space Euler number can be expressed in the matrix form as

$$
e = \frac{2\pi}{A}\text{Tr}_{\boldsymbol{k}}\text{Pf}_{\text{occ}}(P_{\boldsymbol{k}}[\partial_{k_x}P_{\boldsymbol{k}}, \partial_{k_y}P_{\boldsymbol{k}}]).
\tag{C.6}
$$

In a translational invariant system, the $\boldsymbol{k}$ space and the coordinate space can be connected via the Fourier transformation. Therefore, we have

$$
\begin{aligned}
\partial_{k_x}\hat{P}(\boldsymbol{k}) &\to \frac{1}{\delta k_x}(P_{\boldsymbol{k}+\delta\mathbf{k}} - P_{\boldsymbol{k}}) \\
&= \frac{1}{\delta k_x}(U_{\boldsymbol{k}+\delta\mathbf{k},\boldsymbol{r}} P U_{\boldsymbol{k}+\delta\mathbf{k},\boldsymbol{r}}^{\dagger} - U_{\boldsymbol{k},\boldsymbol{r}} P U_{\boldsymbol{k},\boldsymbol{r}}^{\dagger}) \\
&= U_{\boldsymbol{k},\boldsymbol{r}}[\frac{1}{\delta k_x}(U_{\delta\mathbf{k},\boldsymbol{r}} P U_{\delta\mathbf{k},\boldsymbol{r}}^{\dagger} - P)]U_{\boldsymbol{k},\boldsymbol{r}}^{\dagger} \,,
\end{aligned}
\tag{C.7}
$$

where $\delta\mathbf{k} = (\delta k_x, 0)$. If we set $\delta k_x = \frac{2\pi}{L_x}$, then $U_{\delta\mathbf{k},\boldsymbol{r}}$ is just the unitary position matrix $U = \mathrm{e}^{\mathrm{i}\frac{2\pi}{L_x}X}$. Similarly, the relation applies to the other unitary position matrix $V = \mathrm{e}^{\mathrm{i}\frac{2\pi}{L_y}Y}$.

Based on these quantities defined in $\boldsymbol{r}$ space, the Euler number in Eq. (C.6) can be reformulated as

$$
\begin{aligned}
e &= \frac{1}{2\pi}\text{Pf}_{\text{occ}}\sum_{\boldsymbol{k}}U_{\boldsymbol{k},r}P[UPU^{\dagger},VPV^{\dagger}]U_{\boldsymbol{k},r}^{\dagger} \\
&= \frac{1}{2\pi}\text{Pf}_{\text{occ}}\text{Tr}(U_{\boldsymbol{k},r}P[UPU^{\dagger},VPV^{\dagger}]U_{\boldsymbol{k},r}^{\dagger}) \\
&= \frac{1}{2\pi}\text{Pf}_{\text{occ}}\text{Tr}(P[UPU^{\dagger},VPV^{\dagger}]),
\end{aligned}
\tag{C.8}
$$

where the last equation holds because of the invariant property of the trace under any unitary transformation. Since the trace and Pfaffian operations act on different individual subspaces, they are commutative as operators on the Wannier basis with $N_{\text{occ}} = 2$, which proves exactly the Eq. (6).

In principle, when $\delta_{k_x}$ is small enough, one can perform the Taylor expansion up to the first order,

$$
\frac{1}{\delta k_x}(U_{\delta\mathbf{k},r}PU_{\delta\mathbf{k},r}^{\dagger} - P) \approx \text{i}[X,P],
\tag{C.9}
$$

to the right side of Eq. (C.7). However, for a $\mathcal{PT}$-symmetric system with real eigenbasis $\{|u_n(\boldsymbol{k})\rangle\}$, both the projection operator $P_{\boldsymbol{k}}$ and its derivative $\partial_{k_x}P_{\boldsymbol{k}}$ are supposed to be real-valued. The first-order expansion term $\text{i}[X,P]$, which deviates from the real field $\mathbb{R}$, should cancel with some other first-order terms (and higher-order terms may contribute significantly) to ensure the real-valued final expression. Therefore, the additional real-value limitation from the $\mathcal{PT}$ symmetry necessitates the use of the unitary position matrix $U$ instead of the usual position matrix $X$ in our final expression of the $\boldsymbol{r}$-space Euler number. This is different from the case of the Chern number where the first-order expansion is applicable to yield a simplified $\boldsymbol{r}$-space formula in Ref. [11, 14].

# D  Numerical implementation of the real-space Euler number

In this section, we demonstrate the practical calculation of Eq. (6) in the main text. We begin by selecting a suitable basis for expressing the operators in the equation. Once this basis is established, we can straightforwardly apply trace and Pfaffian operations.

We initially work with a set of local coordinate space bases, from which we construct diagonal matrices representing the unitary position operators $\hat{U}$ and $\hat{V}$. The projector $\hat{P}$ is defined as

$$
\mathbf{1}_{occ} = \begin{pmatrix} \mathbf{0} & \mathbf{0} \\ \mathbf{0} & \mathbf{1} \end{pmatrix},
\tag{D.1}
$$

in the eigenbasis of the Hamiltonian, with eigenvalues arranged in descending order. Here, $\mathbf{0}$ and $\mathbf{1}$ represent the null matrix and identity matrix, respectively.

To proceed, we diagonalize the Hamiltonian to obtain the eigenvalues and eigenvectors in the local basis. This allows us to create a unitary transformation matrix from the local basis to the eigenbasis of the system. In other words, we have

$$
H = \Pi D\Pi^{-1},
\tag{D.2}
$$

where $D$ is a diagonal matrix with the eigenvalues in descending order, and the columns of $\Pi$ are the corresponding eigenvectors. Subsequently, we determine the explicit expression of the projector $\hat{P}$ through this unitary transformation of the basis, as follows:

$$
P = \Pi\mathbf{1}_{occ}\Pi^{-1}.
\tag{D.3}
$$

All operators are now represented in a unified local basis, simplifying the matrix calculations. To carry out the trace and Pfaffian operations, a basis transformation from the initial local basis to a composite Wannier basis is required. This Wannier basis can be constructed from the eigenbasis by minimizing the Marzari-Vanderbilt localization functional [84,85]. Once we have the transformation matrix $\Pi$ from the eigenbasis to the local basis and $S$ from the eigenbasis to the composite Wannier basis, we can obtain the matrix form of the expression within the brackets in Eq. (C.8):

$$M = S\Pi^{-1}P[UPU^{\dagger}, VPV^{\dagger}]\Pi S^{-1}. \tag{D.4}$$

In this basis, the matrix entries are denoted as $M_{n'n'',r'r''}$. Then the trace operation simply involves summing over the coordinate index $r$, expressed as

$$\mathrm{Tr} := \sum_{r',r''} \delta_{r',r''}. \tag{D.5}$$

Finally, the $r$-space Euler number can be obtained by performing the Pfaffian over occupied space as[1]

$$\mathrm{Pf}_{\mathrm{occ}}(\mathrm{Tr}M) = \mathrm{Pf}(\mathrm{Tr}M)_{\mathrm{occ}}. \tag{D.6}$$

The final step of basis transformation is crucial for accurately calculating the $r$-space Euler number. This transformation is necessary because only on the Wannier basis can we effectively separate the total space into internal and coordinate spaces. When using a set of local basis functions with high localization properties, such as atomic orbitals, the hopping terms of the Hamiltonian naturally mix the coordinate and internal spaces. As a result, it becomes challenging to distinguish the occupied subspace within the internal space, making it difficult to perform the Pfaffian operation using this basis. On the other hand, the eigenbasis of the Hamiltonian is not suitable either. Although it allows for the easy identification of the occupied subspace, this highly delocalized basis presents difficulties in aligning it in a meaningful way to perform the trace and Pfaffian operations correctly.

# E  The distinction between the real-space Chern and Euler numbers

In this section, we give some remarks on the distinction between the real-space Chern and Euler numbers. First, the analysis we've conducted can be directly applied to the Chern class, and the resultant $r$-space expression is nothing but the Bott index,

$$\mathrm{Bott}(\hat{U}, \hat{V}) = \frac{1}{2\pi}\mathrm{Im}\mathrm{Tr}\log(\hat{U}\hat{V}\hat{U}^{-1}\hat{V}^{-1}), \tag{E.1}$$

with $\hat{U} = \hat{P}\exp(2\pi i\hat{X}/L_x)\hat{P}$ and $\hat{V} = \hat{P}\exp(2\pi i\hat{Y}/L_y)\hat{P}$, which measures the commutativity of the position operators and offers an identical topological classification as the Chern number [14,15]. The Bott index can be further simplified by applying the Taylor expansion of the unitary position operator up to the first order, which yields the conventional $r$-space formula of the Chern number in Ref. [11,14]

$$c_1 = \frac{4\pi}{L^2}\mathrm{Im}\mathrm{Tr}'(\hat{P}[\hat{X},\hat{P}][\hat{Y},\hat{P}]), \tag{E.2}$$

where $\hat{X}, \hat{Y}$ are the usual position operators and $\mathrm{Tr}'$ is the usual trace operation acting on the whole space, distinguished from the aforementioned Tr acting on coordinate subspace only.

---

[1]The package code is available at https://github.com/li-dexin-phy/realeulernum.

However, there are significant differences between the $r$-space formulation of the Euler defined in Eq. (6) and Chern number. This distinction arises because the Chern and Euler classes are defined by distinct invariant polynomials of the curvature [82]. When calculating the Chern number in real space, the trace operation is applied to both the internal and coordinate spaces, resulting in a simplified expression with only a single trace operation. In contrast, when calculating the $r$-space Euler number, it becomes essential to distinguish between the coordinate space and the internal space, which requires trace and Pfaffian operations, respectively.

The discussion is more clear in the frame of matrix form. For any operator of the form $M = \begin{pmatrix} \mathbf{0} & \mathbf{0} \\ \mathbf{0} & M_{occ} \end{pmatrix}$ with $\mathbf{0}$ being the null matrix, the relation $\mathrm{Tr}_{occ} M := \mathrm{Tr} M_{occ} = \mathrm{Tr} \begin{pmatrix} \mathbf{0} & \mathbf{0} \\ \mathbf{0} & M_{occ} \end{pmatrix}$ always holds. This is because the trace operation is just to sum over the diagonal of the matrix $M$, which means that the trace over a specific matrix is equal to the trace over the direct sum of this matrix and any null matrix. Therefore, we can safely consider the whole space without further restriction in the occupied space and the result remains the same. However, the Pfaffian does not possess this property, i.e., $\mathrm{Pf}(\begin{pmatrix} \mathbf{0} & \mathbf{0} \\ \mathbf{0} & M_{occ} \end{pmatrix}) = \mathbf{0}$. What's more, the ordering of the basis does not matter for the trace since the sum operation is commutative, while the ordering is crucial in the definition of the Pfaffian. Therefore, although a single $\mathrm{Tr}'$ is enough for calculating the $r$-space Chern number, it is important to find such a basis that can distinguish the internal space from the coordinate space.

This distinction is already evident in the $k$-space scenario. In a periodic lattice, the Bloch states $\{|\psi_n(k)\rangle\}$ can be transformed into Wannier states, which inherently distinguish the coordinate space from the internal space. Specifically, in such a translational invariant system, the Hamiltonian commutes with the translation operator, indicating a common eigenvalue for both operators. Since the energy index $n$ and $k$ denoting quasi-momentum are independent of each other, it is straightforward to change the basis of $k$ via the Fourier transformation to $r$ without mixture from $n$ and derive the Wannier basis. However, if the system lacks translational invariance, the usual Fourier transformation from Bloch states fails to generate Wannier states. Consequently, it becomes crucial to consider composite Wannier functions defined in real space via a unitary transformation from energy eigenstates, without imposing further restrictions.

Secondly, It is worth noting that there is a gauge freedom in the Wannier functions and the determination of the exponentially localized Wannier functions is significant [84]. The existence of the nontrivial Euler number prohibits finding such a basis of Wannier functions, which means that in a space-time inversion symmetric two-dimensional system, the exponentially localized Wannier functions can not be constructed in a phase with nontrivial Euler number [49]. Nevertheless, this is not an obstacle to search for the required composite Wannier functions that are not exponentially localized [88].

# F  Averaging the local Euler marker in finite systems with OBC

In finite systems with OBC, a striking contrast emerges between the local Chern marker and the local Euler marker. While averaging the local Chern marker over such systems yields vanishing results, the same averaging process for the local Euler marker results in non-vanishing values. This disparity highlights a fundamental distinction between the Chern number and the Euler number when calculated in finite systems under OBC, as elaborated below.

To calculate the $r$-space Chern number $c_1$ in Eq. (E.2), we employ standard position operators $\hat{X}$ and $\hat{Y}$ to construct the operator $\hat{P}[\hat{X},\hat{P}][\hat{Y},\hat{P}]$. Notably, the imaginary part of this operator is directly proportional to $c_1$ when subjected to a trace operation [11, 14]:

$$c_1 \propto \mathrm{Im}\mathrm{Tr}'(\hat{P}[\hat{X},\hat{P}][\hat{Y},\hat{P}]) \,. \tag{F.1}$$

Utilizing the transpose invariance and the cyclic property of the trace operation and considering the symmetry of operators $\hat{X}$ and $\hat{Y}$, we can rigorously demonstrate the vanishing of the $r$-space Chern number under OBC [14]:

$$
\begin{aligned}
c_1 \propto &\mathrm{Im}\mathrm{Tr}'(\hat{P}[\hat{X},\hat{P}][\hat{Y},\hat{P}]) \\
=&\mathrm{Im}\mathrm{Tr}'(\hat{P}(\hat{X}\hat{P}-\hat{P}\hat{X})(\hat{Y}\hat{P}-\hat{P}\hat{Y})) \\
=&\mathrm{Im}\mathrm{Tr}'(\hat{P}\hat{X}\hat{P}\hat{Y}\hat{P}-\hat{P}\hat{X}\hat{P}^2\hat{Y}-\hat{P}^2\hat{X}\hat{Y}\hat{P}+\hat{P}^2\hat{X}\hat{P}\hat{Y}) \\
=&\mathrm{Im}\mathrm{Tr}'(\hat{P}\hat{X}\hat{P}\hat{Y}\hat{P}-\hat{P}\hat{X}\hat{P}\hat{Y}-\hat{P}\hat{X}\hat{Y}\hat{P}+\hat{P}\hat{X}\hat{P}\hat{Y}) \\
=&\mathrm{Im}\mathrm{Tr}'(\hat{P}\hat{X}\hat{P}\hat{Y}\hat{P}-\hat{P}\hat{X}\hat{Y}\hat{P}) \,,
\end{aligned}
\tag{F.2}
$$

where we utilize the property of the projection operator, $\hat{P}^2 = \hat{P}$. Note that $\hat{P}$, $\hat{X}$ and $\hat{Y}$ are all Hermitian, we can further simplify $c_1$ by expending the imaginary part as the subtract of the operator with its conjugate,

$$
\begin{aligned}
c_1 \propto &\frac{1}{2i}(\mathrm{Tr}'(\hat{P}\hat{X}\hat{P}\hat{Y}\hat{P}-\hat{P}\hat{X}\hat{Y}\hat{P})-\mathrm{Tr}'(\hat{P}\hat{X}\hat{P}\hat{Y}\hat{P}-\hat{P}\hat{X}\hat{Y}\hat{P})^*) \\
=&\frac{1}{2i}(\mathrm{Tr}'(\hat{P}\hat{X}\hat{P}\hat{Y}\hat{P}-\hat{P}\hat{X}\hat{Y}\hat{P})-\mathrm{Tr}'(\hat{P}\hat{X}\hat{P}\hat{Y}\hat{P}-\hat{P}\hat{X}\hat{Y}\hat{P})^\dagger) \\
=&\frac{1}{2i}(\mathrm{Tr}'(\hat{P}\hat{X}\hat{P}\hat{Y}\hat{P}-\hat{P}\hat{X}\hat{Y}\hat{P})-\mathrm{Tr}'(\hat{P}\hat{Y}\hat{P}\hat{X}\hat{P}-\hat{P}\hat{Y}\hat{X}\hat{P})) \,.
\end{aligned}
\tag{F.3}
$$

This relationship is established through the transpose invariance of the trace operation, i.e.,

$$\mathrm{Tr}'\hat{A} = \mathrm{Tr}'\hat{A}^{\mathrm{T}} \,, \tag{F.4}$$

which leads to

$$\mathrm{Im}\mathrm{Tr}'\hat{A} = \frac{1}{2i}(\mathrm{Tr}'\hat{A}-\mathrm{Tr}'\hat{A}^*) = \frac{1}{2i}(\mathrm{Tr}'\hat{A}-\mathrm{Tr}'\hat{A}^\dagger) \,. \tag{F.5}$$

Then, using the well-known cyclic property of trace operation, i.e., for general matrices $\hat{A}$ and $\hat{B}$, it is known that

$$\mathrm{Tr}'(\hat{A}\hat{B}) = \mathrm{Tr}'(\hat{B}\hat{A}) \,, \tag{F.6}$$

$c_1$ can be further simplified as

$$
\begin{aligned}
c_1 \propto &\frac{1}{2i}(\mathrm{Tr}'(\hat{P}\hat{X}\hat{P}\hat{Y}\hat{P})-\mathrm{Tr}'(\hat{P}\hat{X}\hat{Y}\hat{P})-\mathrm{Tr}'(\hat{P}\hat{Y}\hat{P}\hat{X}\hat{P})+\mathrm{Tr}'(\hat{P}\hat{Y}\hat{X}\hat{P})) \\
=&\frac{1}{2i}(\mathrm{Tr}'(\hat{X}\hat{P}\hat{Y}\hat{P}^2)-\mathrm{Tr}'(\hat{X}\hat{P}^2\hat{Y}\hat{P})-\mathrm{Tr}'(\hat{X}\hat{Y}\hat{P}^2)+\mathrm{Tr}'(\hat{Y}\hat{X}\hat{P}^2)) \\
=&\frac{1}{2i}(\mathrm{Tr}'(\hat{X}\hat{P}\hat{Y}\hat{P})-\mathrm{Tr}'(\hat{X}\hat{P}\hat{Y}\hat{P})-\mathrm{Tr}'(\hat{X}\hat{Y}\hat{P})+\mathrm{Tr}'(\hat{Y}\hat{X}\hat{P}) \\
=&-\frac{1}{2i}\mathrm{Tr}'(\hat{X}\hat{Y}\hat{P}-\hat{Y}\hat{X}\hat{P}) \\
=&-\frac{1}{2i}\mathrm{Tr}'([\hat{X},\hat{Y}]\hat{P}) \\
=&0 \,,
\end{aligned}
\tag{F.7}
$$

where we have used the the symmetry of operators $\hat{X}$ and $\hat{Y}$

$$[\hat{X},\hat{Y}] = 0 \,. \tag{F.8}$$

In summary, the vanishing of the $r$-space Chern number under OBC arises from a cancellation effect, driven by three key factors:

- Transpose invariance of the trace operation: $\mathrm{Tr}'\hat{A} = \mathrm{Tr}'\hat{A}^{\mathrm{T}}$.

- Cyclic property of the trace operation: $\mathrm{Tr}'(\hat{A}\hat{B}) = \mathrm{Tr}'(\hat{B}\hat{A})$.

- Symmetry of standard position operators $\hat{X}$ and $\hat{Y}$: $[\hat{X}, \hat{Y}] = 0$.

In contrast, calculating the $r$-space Euler number doesn't encounter a similar cancellation effect, primarily due to the distinct properties of the trace operation and the Pfaffian. First, the transpose invariance, which holds for the trace operation, does not apply to the Pfaffian. For a general skew-symmetric matrices $\hat{A}$, we have

$$\mathrm{Pf}\hat{A}^{\mathrm{T}} = \mathrm{Pf}(-\hat{A}) = \pm\mathrm{Pf}\hat{A}, \tag{F.9}$$

with the additional sign depending on $N_{\mathrm{occ}}$. Second, unlike the trace operation, the Pfaffian lacks the necessary cyclic properties for straightforward cancellation,

$$\mathrm{Pf}(\hat{A}\hat{B}) \neq \mathrm{Pf}(\hat{B}\hat{A}). \tag{F.10}$$

To be more specific, we now defined a $r$-space quantity $\zeta$ with trace operation only as

$$\begin{aligned}
\zeta &= \frac{1}{2\pi}\mathrm{Tr}'(\hat{P}[\hat{P}_U, \hat{P}_V]) \\
&= \frac{1}{2\pi}\mathrm{Tr}'(\hat{P}\hat{P}_U\hat{P}_V - \hat{P}\hat{P}_V\hat{P}_U),
\end{aligned} \tag{F.11}$$

where $\hat{P}_U = \hat{U}\hat{P}\hat{U}^\dagger$ and $\hat{P}_V = \hat{V}\hat{P}\hat{V}^\dagger$ are defined analogous to the expression of $r$-space Euler number.

Since the projector $\hat{P}$ is Hermitian and $\hat{U}/\hat{V}$ are both unitary operators, it is easy to prove that both operators are Hermitian operators. In addition, we further express the unitary position operators $\hat{U}^\dagger$ and $\hat{V}^\dagger$ as

$$\begin{aligned}
\hat{U}^\dagger &= \hat{U}^{-1} = \exp(2\pi\mathrm{i}(-\hat{X})/L_x) = \hat{I}\hat{U}\hat{I}, \quad \text{and} \\
\hat{V}^\dagger &= \hat{V}^{-1} = \exp(2\pi\mathrm{i}(-\hat{Y})/L_Y) = \hat{I}\hat{V}\hat{I},
\end{aligned} \tag{F.12}$$

where $\hat{I}$ is the inversion operator. And in $\mathcal{PT}$-symmetric system, the projector $\hat{P}$ is invariant under such inversion.

Since both Hamiltonian $\hat{H}$ and projector $\hat{P}$ satisfy the reality condition, the operators share the transpose invariant property as

$$\begin{aligned}
\hat{P}^T &= \hat{P}^{*\dagger} = \hat{P}^\dagger = \hat{P}, \\
\hat{U}^T &= \hat{U}, \\
\hat{V}^T &= \hat{V}, \\
\hat{P}_U^T &= \hat{U}^\dagger\hat{P}\hat{U}, \quad \text{and} \\
\hat{P}_V^T &= \hat{V}^\dagger\hat{P}\hat{V}.
\end{aligned} \tag{F.13}$$

Now we can obtain the equivalent form of the first term in Eq. (F.11) as

$$\begin{aligned}
\frac{1}{2\pi}\mathrm{Tr}'(\hat{P}\hat{P}_U\hat{P}_V) &= \frac{1}{2\pi}\mathrm{Tr}'(\hat{P}\hat{P}_U\hat{P}_V)^T = \frac{1}{2\pi}\mathrm{Tr}'(\hat{P}_V^T\hat{P}_U^T\hat{P}) = \frac{1}{2\pi}\mathrm{Tr}'((\hat{V}\hat{P}\hat{V}^\dagger)^T\hat{P}_U^T\hat{P}) \\
&= \frac{1}{2\pi}\mathrm{Tr}'(\hat{V}^\dagger\hat{P}\hat{V}\hat{P}_U^T\hat{P}) = \frac{1}{2\pi}\mathrm{Tr}'(\hat{I}\hat{V}\hat{I}\hat{P}\hat{I}\hat{V}^\dagger\hat{I}\hat{P}_U^T\hat{P}) = \frac{1}{2\pi}\mathrm{Tr}'(\hat{I}\hat{P}_V\hat{I}\hat{P}_U^T\hat{P}) = \frac{1}{2\pi}\mathrm{Tr}'(\hat{I}\hat{P}_V\hat{P}_U\hat{P}\hat{I}) \\
&= \frac{1}{2\pi}\mathrm{Tr}'(\hat{P}\hat{P}_V\hat{P}_U),
\end{aligned} \tag{F.14}$$

which is just the second term of Eq. (F.11). Therefore, we prove that $\zeta$ is trivial. Again, we notice that Eq. (F.4) and Eq. (F.5) are used in the first and the last equality respectively.

However, in the case of $r$-space Euler number $e$, as we have already discussed via Eq. (F.9) and Eq. (F.10), such cancellation doesn't exist. Hence, it becomes possible to calculate the $r$-space Euler number under OBC.

# G  Brief discussion of the reality condition in $\mathcal{PT}$-broken systems

Although we focus on the $\mathcal{PT}$-symmetric system in the main text, it is not a constraint on calculating the $r$-space Euler number. In $k$ space, since the time reversal $\mathcal{T}$ can be considered a conjugate operator combined with a unitary matrix and a sign flip of $k$, a $\mathcal{T}$-invariant Hamiltonian $H(k)$ satisfies $H(k) = \hat{T}H(k)\hat{T}^{-1} = H^\star(-k)$ under a proper basis obtained from Takagi decomposition. Therefore, only in a few time-reversal invariant momenta with $k = -k$ can we derive a real Hamiltonian. To keep the Hamiltonian real in the whole $k$-space, another operator such as $\mathcal{P}$ and $\mathcal{C}_{2z}$ that can reverse the sign of $k$ is essential. However, in $r$ space, the time reversal $\mathcal{T}$ no longer acts on the sign of $k$. This means that the symmetry requirement for the reality condition is only the time reversal $\mathcal{T}$. Therefore, in a finite system with OBC lacking spatial symmetry, we can again obtain the necessary basis from the initial local basis via a transformation matrix given by the Takagi decomposition. Under the new basis, the Hamiltonian is real-valued. By solving the eigenvalue problem of the Hamiltonian in such basis, the transformation matrix constructed by all eigenfunctions of the Hamiltonian is real-valued as well. This is just the reality condition necessitated for the definition of the Euler class. Consequently, one can apply the real-space formula of the Euler number to any nonmagnetic aperiodic systems with open boundary, such as quasicrystals, and amorphous materials without any spatial symmetries.

# H  Details of the model and method

## H.1  Model

All the calculations are performed based on the tight-binding Hamiltonian in Eq. (8). The hopping integral $t_{\mu\nu}(r_{ij})$ follows the Slater-Koster parameterization which depends on the orbital type and the directional cosines of the intersite vector $r_{ij} = r_i - r_j$. To be more specific, the explicit expression of Slater-Koster parameterized hopping integral are listed:

$$t_{p_x,p_x} = l^2 V_{pp\sigma} + (1-l^2)V_{pp\pi},$$
$$t_{p_y,p_y} = m^2 V_{pp\sigma} + (1-m^2)V_{pp\pi},$$
$$t_{p_x,p_y} = lm(V_{pp\sigma} - V_{pp\pi}),$$
$$t_{d_{x^2-y^2},d_{x^2-y^2}} = \frac{3}{4}V_{dd\sigma}(l^2-m^2)^2 + (l^2+m^2-(l^2-m^2)^2)V_{dd\pi} + \frac{1}{4}(l^2-m^2)^2 V_{dd\delta},$$
$$t_{d_{xy},d_{xy}} = l^2 m^2(3V_{dd\sigma} - 4V_{dd\pi} + V_{dd\delta}) + V_{dd\pi},$$
$$t_{d_{xy},d_{x^2-y^2}} = \frac{3}{2}lm(l^2-m^2)V_{dd\sigma} + 2lm(m^2-l^2)V_{dd\pi} + \frac{1}{2}lm(l^2-m^2)V_{dd\delta},$$
$$t_{p_x,d_{x^2-y^2}} = \frac{\sqrt{3}}{2}l(l^2-m^2)V_{pd\sigma} + l(1-l^2+m^2)V_{pd\pi},$$
$$t_{p_y,d_{x^2-y^2}} = \frac{\sqrt{3}}{2}m(l^2-m^2)V_{pd\sigma} - m(1+l^2-m^2)V_{pd\pi},$$

$$
\begin{aligned}
t_{p_x,d_{xy}} &= \sqrt{3}l^2 m V_{pd\sigma} + (1-2l^2)m V_{pd\pi}\,, \\
t_{p_y,d_{xy}} &= \sqrt{3}lm^2 V_{pd\sigma} + (1-2m^2)l V_{pd\pi}\,, \\
t_{p_y,p_x} &= t_{p_x,p_y}\,, \\
t_{d_{x^2-y^2},d_{xy}} &= t_{d_{xy},d_{x^2-y^2}}\,, \\
t_{d_{x^2-y^2},p_x} &= -t_{p_x,d_{x^2-y^2}}\,, \\
t_{d_{x^2-y^2},p_y} &= -t_{p_y,d_{x^2-y^2}}\,, \\
t_{d_{xy},p_x} &= -t_{p_x,d_{xy}}\,, \\
t_{d_{xy},p_y} &= -t_{p_y,d_{xy}}\,,
\end{aligned}
\tag{H.1}
$$

where $\hat{r}_{ij} = (l,m)$ is the unit direction vector. The hopping strength is chosen to have an inverse-square decay with the distance as $t_{\mu\nu}(r_{ij}) \propto |r_{ij}|^{-2}$. We adopt the equilibrium interatomic bond length as the unit length $a$ of the systems, which is the lattice constant for the perfect square lattice and the side length of basic building blocks (square and rhombus) for the Ammann-Beenker-tiling quasicrystals. In numerical calculations, we set the unit length of the system $a=1$ for simplicity.

We consider a 2D square lattice with a band inversion at the $\Gamma$-point in $\boldsymbol{k}$-space between degenerate $p_{x,y}$ and $d_{x^2-y^2,xy}$ orbitals, as shown in Fig. 1(a). In real space, we investigate $L \times L$ supercells of the square lattice with periodic boundary condition (PBC) or open boundary condition (OBC). For convenience, we choose the lattice size $L$ to be an odd integer, which allows the supercell to possess an inversion center located at its central site.

## H.2 Disorder of on-site energy

The tight-binding Hamiltonian with the onsite disorder is under our consideration as well. Therefore, we introduce a disorder term to the Hamiltonian H as

$$
H(\{\lambda_i\}) = H + \sum_{i\mu} \lambda_i c_{i\mu}^\dagger c_{i\mu}\,,
\tag{H.2}
$$

where $\{\lambda_i\}$ is a set of random on-site energy added to one-half sites of the whole sample. Here $\{\lambda_i\}$ distribute uniformly within the interval of $[-W, W]$ with $W$ being the disorder strength. To preserve the inversion symmetry, the on-site energies of the rest sites of the sample are determined by inversion. Namely, each pair of sites connected by the inversion symmetry shares the same on-site energy. The calculations are performed in samples with lattice size $L = 31$. Because of the random character, we average the $\boldsymbol{r}$-space Euler number over 100 sample configurations for every $W$. A higher accuracy can be achieved by adopting samples with larger sizes and doing the statistical average for more samples.

## H.3 Structural disorder

In order to further investigate the applicability of the real-space formula of the Euler number, we study the effect of in-plane structural disorder in finite samples which lack the translational periodicity and all other spatial symmetries [101–103]. To illustrate this effect, we assign random atomic displacement $\boldsymbol{\delta} = (d\cos\theta, d\sin\theta)$ away from its equilibrium position for each atom of the aforementioned 2D perfect square lattice, as depicted in Fig. S7(a). Here, $\theta$ is a random azimuth angle uniformly distributed in the interval $[0, 2\pi)$. The amplitude $d$ of atomic displacements are determined by Gaussian distributions with standard deviation $\sigma = 0.2$. For the special structural disorder case but preserving the inversion symmetry, one can assign the random atomic displacement only to the first half of the lattice, and determine the locations

of atoms in the other half of the lattice by the inversion symmetry. As the structure becomes disordered, the hopping integrals in Eq. (8) also adjust according to local structural distortions.

### H.4 Twisted boundary condition for quasicrystals

For an octagonal sample of the Ammann-Beenker-tiling quasicrystal, we calculate the energy spectrum using both OBC and the twisted boundary condition (TBC). To apply TBC, we artificially glued the opposite edges of an octagonal polygon. Specifically, for an octagonal polygon with the edge width of $L_{edge}$, we label the edges as $E_p$ ($p = 1, 2, \cdots 8$) anticlockwise. For the edge $E_p$, we define a translation operator, which is perpendicular to the edge and translates the octagon by a distance of $2L_{edge}$. By applying the translation operator to the finite octagonal quasicrystal so that edge $E_p$ of the sample connects with the opposite edge $E_{(p+4) \mod 8}$ of the translated image sample. Then we consider the hopping cross the edge between site $i$ in the octagonal sample and site $\tilde{j}$ in the image sample. These extra hoppings also follow the Slater-Koster parameterization and have inverse-square decay with the distance (i.e., $|\boldsymbol{r}_{i\tilde{j}}|^{-2}$). Therefore, in addition to the intersite hoppings between sites inside the sample, we also consider extra hoppings between sites near opposite edges. Importantly, by applying TBC, we not only get rid of the effect of the open boundary but also restore the 8-fold symmetry of the quasicrystal.

### H.5 Construction of composite Wannier function

Since the real-space Euler number obtained in Eq. (6) can only be calculated in Wannier basis, a crucial step in the numerical calculation is to construct the Wannier function in systems even without the spatial translational symmetry.

One possible way to construct the real-space Wannier function is the functional optimization method. The eigenfunctions $\phi_m$ associated with the energy index $m$ can be obtained by solving the eigenvalue problem of the Hamiltonian $H$. Then the required composite Wannier functions $W_n$ are constructed from $\phi_m$ as

$$W_n = \sum_m S_{nm} \phi_m, \tag{H.3}$$

via the unitary transformation $S$ that can be considered as the combination of a phase term and a band matrix [93], which can be numerically obtained by minimizing the Wannier spread functional

$$\Omega = \sum_n \left[ \langle W_n | r^2 | W_n \rangle - \langle W_n | r | W_n \rangle^2 \right]. \tag{H.4}$$

Once the Wannier functions are constructed, the internal and coordinate spaces can be easily separated and the real-space Euler number can be calculated straightforwardly using the formula given in Eq. (6).

In the functional optimization process, a key factor is the selection of the initialization. In our case, this is the initial value of $W_n$. In principle, the initial Wannier function can be set arbitrarily. However, to obtain a more efficient and convergent result, the initial function can be chosen as the Wannier function obtained in a translational invariant system. To be more specific, for disordered lattices, the Wannier function constructed through the Fourier transformation of the Block functions obtained in perfect lattice is a great initial function. However, it might be hard to find such a $\boldsymbol{k}$-space analog in quasicrystal and even totally amorphous systems. In our work, the initial Wannier function of the quasicrystal is obtained from that of a 16×73 rectangle lattice in Fig. 2.

As for the fully non-periodic systems where the gauge optimization fails to work, other methods such as the Iterated projected position (IPP) algorithm are supposed to be considered without the initialization requirement [104].

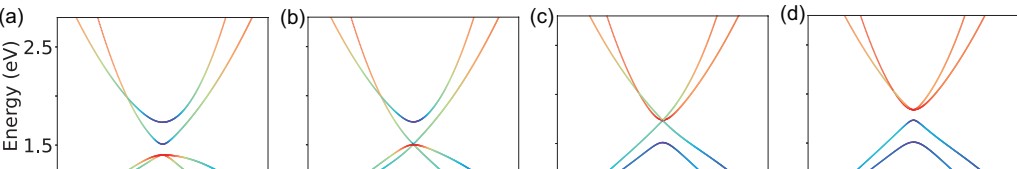

Figure S1: The evolution of band structure around the phase transition in Fig. 1(b). Orbital-resolved band structures near $\Gamma$ point for the square lattice based on Eq. (8) with different on-site energy difference $\Delta$. (a) $\Delta = 6.76$ eV (region I, $e = 1$). (b) $\Delta = \Delta_1 = 6.86$ eV (the critical point between region I and II). (c) $\Delta = \Delta_2 = 7.10$ eV (the critical point between region II and III). (d) $\Delta = 7.20$ eV (region III, $e = 0$).

Another issue to be clarified is the ordering of occupied states within a certain cell $\mathfrak{r}$. It can be determined by the corresponding diagonal element of the Hamiltonian on the composite Wannier basis. We also noticed that the local Euler marker is attached to only an additional minus sign when this ordering is inverted. Therefore, more conveniently, the sign of local Euler markers can be set to satisfy the continuity of these markers.

### H.6 Numerical calculation of the k-space Euler number

Generally speaking, non-accidental degenerate points (nodes) between the nontrivial occupied bands are common in $\boldsymbol{k}$-space [47]. To numerically calculate the $\boldsymbol{k}$-space Euler number in this context, we employ the following expression:

$$|e| = \int_{\mathcal{D}} e(\mathcal{F}) - \int_{\partial \mathcal{D}} \langle u_1 | \nabla | u_2 \rangle \cdot \frac{\mathrm{d}\boldsymbol{k}}{2\pi}, \qquad \text{(H.5)}$$

where $e(\mathcal{F}) = (1/\pi)\langle \partial_{[k_x} u_1(\boldsymbol{k}) | \partial_{k_y]} u_2(\boldsymbol{k}) \rangle \mathrm{d}k_x \wedge \mathrm{d}k_y$, and $\mathcal{D}$ represents the region in the Brillouin zone (BZ) containing those nodes.

## I  More numerical results

### I.1  Band structures around the topological phase transition in Fig. 1(b)

Here we discuss three regions presented in Fig. 1(b) in the main text in detail. These regions are divided by two critical points $\Delta_1 = 6.86$ and $\Delta_2 = 7.10$ eV. As illustrated in Fig. S1(a), there is initially a double band inversions occurring around $\Gamma$ point with $\Delta < \Delta_1$, which accounts for the nontrivial band topology with $|e| = 1$. This is consistent with the calculations of the $\boldsymbol{r}$-space Euler number in the main text, demonstrating that the phase in region I is indeed the Euler insulator.

As the onsite difference $\Delta$ increases, the gap decreases gradually and eventually closes at $\Delta_1$, as shown in Fig. S1(b). The closing of the gap indicates a topological phase transition. However, unlike the usual situation of a single band inversion where the gap reopens immediately after closure accompanied by a sharp change in the topological invariant, our model has an intermediate gapless region before the gap reopens at $\Delta_2$ as shown in Fig. S1(c). From the perspective of the band topology, region II is a one-band-inverted phase without protection from the Euler topology, which accounts for the continuous decreasing of the $\boldsymbol{r}$-space Euler number in region II [see Fig. 1(b) in the main text]. In addition, the distinction between the $\boldsymbol{k}$-space and $\boldsymbol{r}$-space Euler number in region II is also due to the closed gap that brings up the discrimination between $\hat{P}$ projected and the well-defined occupied states. When $\Delta > \Delta_2$

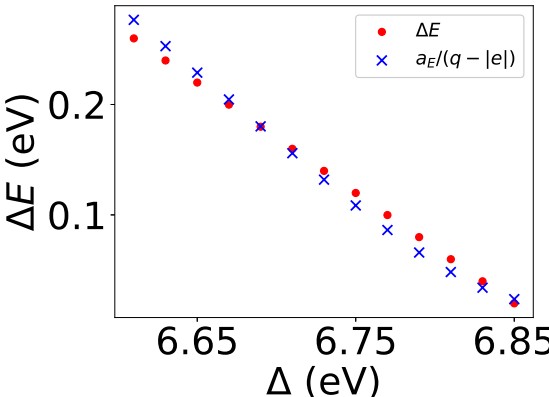

Figure S2: The on-site energy difference $\Delta$ dependence of $\Delta E$ and $q - |e|$ with $a_E = 3.125$ meV. Here q denotes the expected quantized value and is equal to 1 in this case.

as shown in Fig. S1(d), the gap reopens and there is no band inversion at $\Gamma$ point anymore. This phase can be adiabatically connected to the atomic limit without gap closure. Therefore, region III is a trivial insulator with $e = 0$ as expected.

### I.2  Convergence of the real-space Euler number with decreasing band gap

There is a numerical deviation of both $\boldsymbol{k}$-space and $\boldsymbol{r}$-space Euler number from an exact integer in regions I and III near the critical points in Fig. 1(b). Here we examine the numerical deviation in region I. As presented in the main text, the $\boldsymbol{r}$-space Euler number equals the exact one within a correction of order $\mathcal{O}(1/(L\Delta E))$ for systems with lattice size $L$ and energy gap $\Delta E$. For a system with fixed lattice size $L$, the numerical correction is inversely proportional to the band gap: $1 - |e| \propto 1/\Delta E$, where $e$ is the $\boldsymbol{r}$-space Euler number. To examine the convergence of our $\boldsymbol{r}$-space formulation as a function of on-site energy difference $\Delta$, we calculate the band gap $\Delta E$ and the $\boldsymbol{r}$-space Euler number in region I for a sample with fixed $L$. As shown in Fig. S2, we plot the $\Delta$ dependence of both $\Delta E$ and $a_E/1 - |e|$, where $a_E = 3.125$ meV is a fitting parameter. The inverse of the numerical correction fits well with $\Delta E$ as expected, indicating that the numerical correction becomes significant near the critical point of the phase transition. Nevertheless, such numerical correction can be diminished by increasing the lattice size.

### I.3  Convergence of the real-space Euler number with increasing lattice size

To examine the convergence of real-space Euler number as a function of lattice size $L$, we further calculate larger systems with the size $L$ up to 90. We consider the pristine lattice with PBC or OBC and a disordered lattice with $W = 1.0$ eV. For the disordered case, we perform an average of the $\boldsymbol{r}$-space Euler number over 10 samples for each $L$. As shown in Fig. S3, the curve of the disordered case has not saturated yet but converges slowly towards the quantized value of 1. To further check the convergent behavior, we perform a fitting (see the fitting line in Fig. S3) to estimate the lattice size for the real-space Euler number to reach the quantized value with the error less than 1%. It is found that the required lattice sizes are $L \approx 355$ for the pristine OBC case and $L \approx 570$ for the disordered case respectively, which are beyond the memory limit of our computational resource. As a comparison, the same estimation for the pristine PBC case without disorder shows that a much smaller lattice size of $L \approx 63$ is required to reach the same accuracy. This is because the energy gap is $\Delta E = 0.469$ eV in the pristine

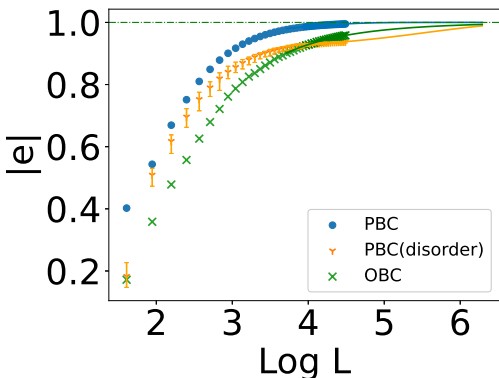

Figure S3: The lattice size $L$ dependence of the $\boldsymbol{r}$-space Euler number calculated without and with on-site energy disorder ($W = 1.0$ eV) using PBC, and without disorder using OBC. Fitting curves are added for each case.

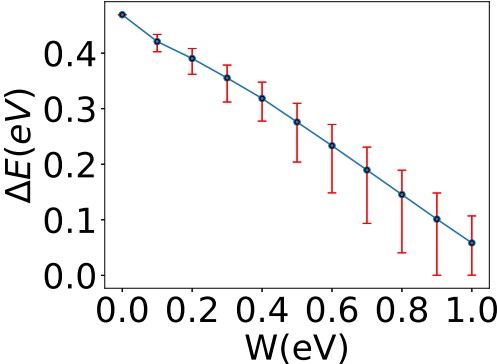

Figure S4: The energy gap $\Delta E$ versus on-site energy disorder strength $W$. For each $W$, the configuration average is performed over 100 realizations of 51×51 disordered lattices with PBC.

PBC case, but for the disordered case with $W = 1.0$ eV, the corresponding averaged gap reduces significantly to $\Delta E = 0.0583$ eV, which is one order of magnitude smaller than the former. This dependence is illustrated in Fig. S4. As the numeric correction is on the order of $\mathcal{O}(1/(L\Delta E))$ for systems with lattice size $L$ and energy gap $\Delta E$, the much slower convergence rate of the disordered case is mainly due to the significant reduction of the energy gap reduction in the presence of disorder.

## I.4 Deviation of real-space Euler number with OBC

In this section, we discuss the deviation of $\boldsymbol{r}$-space Euler number with OBC. The OBC case shows a similar linear dependence between $1/L$ and the numerical deviation $\Delta e = 1-|e|$ with slower convergent behavior. This means that the OBC includes an additional effect which is up to order $\mathcal{O}(1/L)$ as well. Notice that the Euler number is obtained by averaging the local Euler markers at all sites. Since the sites far from boundaries are supposed to preserve similar properties to those in periodic systems, such deviation originates from the sites close to the boundary, which contributes $\mathcal{O}(L_{edge}/A) = \mathcal{O}((L^2 - (L-2)^2)/L^2) = \mathcal{O}(4L/L^2) = \mathcal{O}(4/L)$ as expected. Here $L_{edge}$ and $A$ are the number of sites in the boundary and the whole sample, respectively. This additional factor accounts for the slope approximated to 1/4 in Fig. S5, confirming the effect on $\boldsymbol{r}$-space Euler number from the edge.

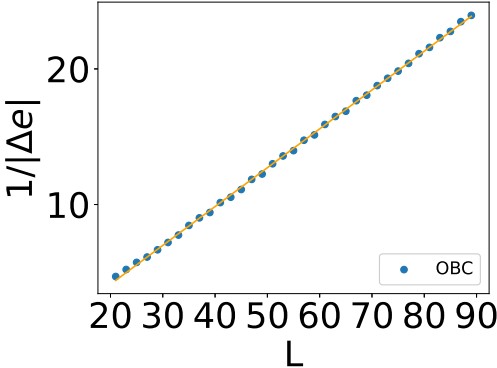

Figure S5: The inverse of the deviation of the $r$-space Euler number $\Delta e$ versus lattice size L with OBC.

## I.5   Local Euler markers in lattices with on-site disorder in Fig. 1(c)

In Fig. 1(c) in the main text, we illustrate another intriguing type of topological phase transition induced by on-site disorder. The averaged $r$-space Euler number $e$ decreases from 1 to 0 with increasing the disorder strength $W$. Here we present the spatial distribution of the local Euler marker of the sample with PBC at different disorder strengths $W$, as shown in Fig. S6. At a relatively weak disorder of $W = 1.2$ eV, the system maintains its nontrivial Euler characteristics. Predominantly, the grid points exhibit nontrivial local Euler markers $e(\mathbf{r}) \approx 1$ with a few isolated points having vanished $e(\mathbf{r}) \approx 0$, as shown in Fig. S6(a). However, by increasing the disorder strength $W$, a noteworthy transformation occurs: the number of trivial points with $e(\mathbf{r}) \approx 0$ increases, and the trivial area enlarges in size, eventually leaving the nontrivial area shrinks to an isolated region in the sample [see Fig. S6(c)]. This isolated nontrivial region with $e(\mathbf{r}) \approx 1$ diminishes in size gradually as $W$ continues to increase, ultimately fragmenting into small segments [see Fig. S6(d,e)]. Upon reaching $W \geq 2.2$ eV, the situation undergoes a significant shift. As shown in Fig. S6(f), none of the grid points exhibits nontrivial local Euler markers, indicating that the system is driven into a trivial phase by strong on-site disorder.

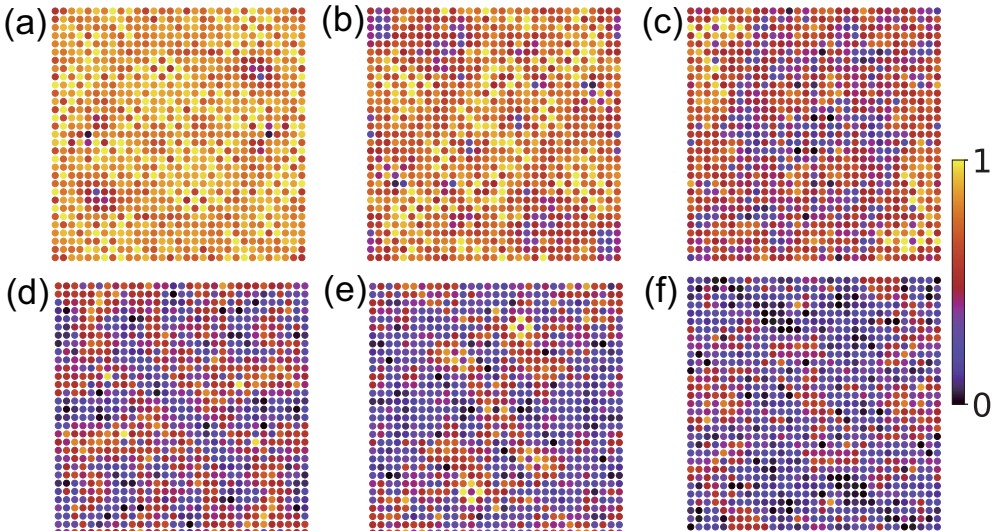

Figure S6: The real-space distribution of local Euler markers $e(\mathbf{r})$ in $31 \times 31$ square lattices with PBC at different disorder strength $W$. (a) $W = 1.2$ eV. (b) $W = 1.4$ eV. (c) $W = 1.6$ eV. (d) $W = 1.8$ eV. (e) $W = 2.0$ eV. (f) $W = 2.2$ eV.

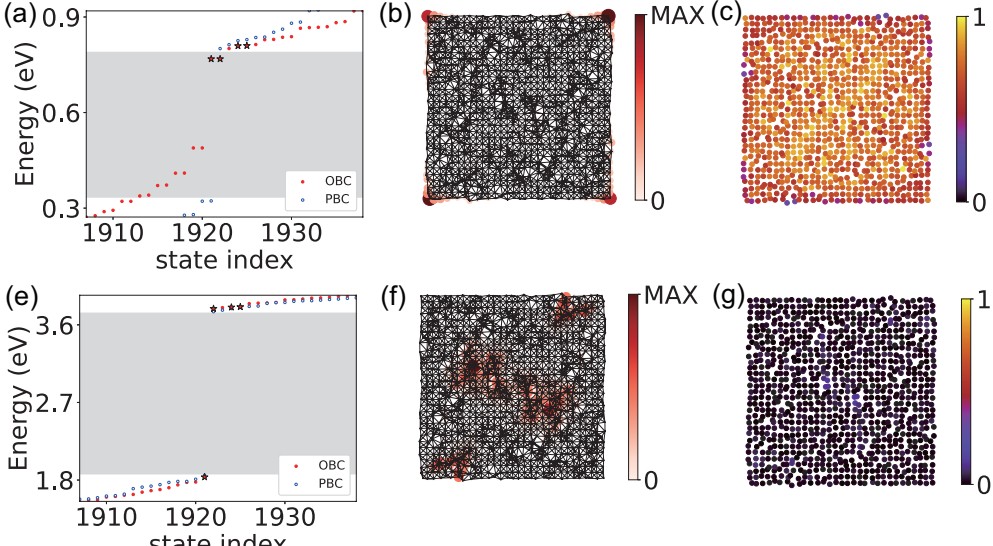

Figure S7: The disordered square lattice model that exhibits band inversions between the $(p_x, p_y)$ and $(d_{x^2-y^2}, d_{xy})$ orbitals. The relevant parameters are as follows: $L = 31$, $\epsilon_{p_x,p_y} = 1.58$, $\epsilon_{d_{x^2-y^2,xy}} = -0.42$, $V_{pp\sigma} = -0.565$, $V_{pp\pi} = -0.044$, $V_{pd\sigma} = 0.773$, $V_{pd\pi} = 0.335$, $V_{dd\sigma} = 0.444$, $V_{dd\pi} = 0.224$, $V_{dd\delta} = 0.659$ eV. (a) The energy eigenvalues versus the state index in the vicinity of the Fermi level for the disordered square lattice with PBC and OBC. (b) The spatial distribution of the corner states [red stars in (a)]. (c) The real-space distribution of the local Euler marker $e(\mathbf{r})$ for the disordered system with OBC. (e-g) Corresponding results as (a-c) for a trivial state with $e = 0$ (The onsite energy difference is set to $\Delta = 6$ eV).

Notably, this type of topological phase transition differs from those in disordered Chern insulators and quantum spin Hall insulators, where a sudden jump of topological invariants occurs at the critical point [25]. Instead, the disorder-induced transition in this model manifests as a more continuous evolution. Physically, we conjecture this to be due to the disorder-induced renormalization of the parameter $\Delta$ which dominates the transition from the Euler insulator to the trivial phase through the intermediate gapless phase, as depicted in Fig. 1(b).

## I.6 Euler insulator in lattice with moderate structural disorder

In this section, we study the Euler topology of a square lattice with moderate structural disorder. We specifically preserve the inversion symmetry at this stage, for comparison with the case breaking all spatial symmetries presented in Fig. 3 of the main text. To construct the structurally disordered square lattice [34, 35, 79, 101–103], we add random displacement $\boldsymbol{\delta} = d(\cos\theta, \sin\theta)$ away from its equilibrium position for each site in one-half sample ($\tau_{1/2}$) of the square lattice, and assign the displacements for the other half to preserve inversion symmetry. Here $\theta$ and $d$ are determined by uniform distributions in the interval $[0, \pi)$ and Gaussian distributions with standard deviation $\sigma = 0.2$, respectively. As shown in Fig. S7(a), the energy spectrum of the structurally disordered lattice with OBC exhibits 4 states at the Fermi level in the bulk gap obtained using PBC (grey area). We plot the spatial distribution of these states and find that they are well localized at 4 corners of the sample [see Fig. S7(b)], implying its higher-order topological feature. Because of the effect of the structural disorder, the corner states move upwards to the bottom of the unoccupied bulk states. Furthermore, we analyze the distribution of the local Euler marker in the finite sample with structural disorder, as shown in Fig. S7(c). The plot confirms that the local Euler markers $e(\mathbf{r})$ are close to the

(a)    (b)

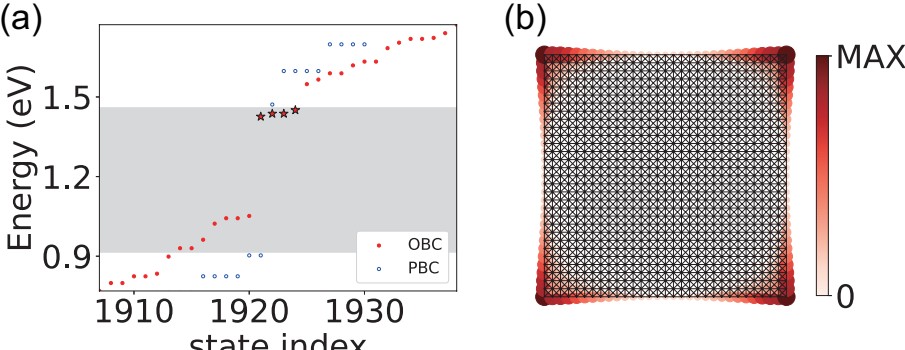

Figure S8: The square lattice model that exhibits band inversions between the $(p_x, p_y)$ and $(d_{x^2-y^2}, d_{xy})$ orbitals. The relevant parameters are as follows: $L = 31$, $\epsilon_{p_x,p_y} = 2.73$, $\epsilon_{d_{x^2-y^2,xy}} = -0.42$, $V_{pp\sigma} = -0.565$, $V_{pp\pi} = -0.044$, $V_{pd\sigma} = 0.773$, $V_{pd\pi} = 0.335$, $V_{dd\sigma} = 0.444$, $V_{dd\pi} = 0.224$, and $V_{dd\delta} = 0.659$ eV. (a) Energy spectrum of the square lattice with PBC and OBC. Four corner states in the gap are highlighted by red stars. (b) Spatial distribution of the corner states [red stars in (a)].

expected value of 1 in the bulk of the sample, while they deviate in the boundary region. As expected, the sum of $e(\mathbf{r})$ over the whole finite sample does not vanish but yields the desired Euler number which should converge to the quantized value with increasing lattice size. Consequently, we can obtain an accurate $\mathbf{r}$-space Euler number by averaging $e(\mathbf{r})$ over an internal region of the sample to get rid of the boundary deviation. As a comparison, we also perform a similar calculation for a trivial phase (see the bottom panels in Fig. S7). As illustrated in Fig. S7(g), the local Euler marker is almost 0 all over the sample, unambiguously indicating the trivial nature of the state.

### I.7 The upward shift of eigenenergies of corner states with decreasing bulk gap

In this section, we discuss the upward shift of eigenenergies of corner states with increasing on-site energy. As illustrated in Fig. 3 and Fig. S7, introducing structural disorder leads to the upward shift of the eigenenergies of corner states. In fact, this effect originates from the decreasing energy gap. In structurally disordered samples, the decrease is attributed to the increasing disorder amplitude. Additionally, the adjustment of the on-site energy can also lead to a smaller bulk gap. As discussed in appendix I.1, in region I, we can lift the on-site energy of $p$-orbitals such that the bulk gap will decrease and finally vanish at critical point $\Delta_1$. Therefore, for comparison purposes, we consider a square model with the same parameters as in Fig. 3 except for the on-site energy difference. As illustrated in Fig. S8(a), increasing the on-site energy difference shows a similar upward shift effect to that observed in structurally disordered lattices. These shifting states near the upper bound of the PBC gap are spatially localized at four corners, as shown in Fig. S8(b). These results show the similarity between the effect of on-site energy difference and structural disorder on the upward shifting, which can be explained as the effect of the decreasing bulk gap.

### I.8 Validation in other models with different Euler numbers

In the main text, we present the results based on the tight-binding model with the Euler number $e = 1$. Now we show that our proposed $\mathbf{r}$-space formula of the Euler number also applies to other models with different Euler numbers as well. Different from the tight-binding Hamiltonian in Eq. (8) based on the atomic orbital basis, we consider a generic $\mathcal{PT}$-symmetric

four-band Bloch Hamiltonian $H^{(\chi_1,\chi_2)}(\boldsymbol{k})$ with $(\chi_1, \chi_2)$ representing the Euler number of the upper and lower two-band subspace respectively [99].

Specifically, we take $(\chi_1, \chi_2) = (2, 2)$ as an example. The time-reversal $\hat{T}$ and inversion $\hat{P}$ operators can be expressed as

$$\hat{T} = -i\Gamma_{22}\hat{K},$$
$$\hat{P} = i\Gamma_{22},$$
(I.1)

where $\Gamma_{i,j} = \sigma_i \otimes \sigma_j$ are $4 \times 4$ Dirac matrices and $\hat{K}$ is the complex conjugation. The minimal four-band Hamiltonian $H^{(2,2)}(\boldsymbol{k})$ can be expressed as

$$H^{(2,2)}(\boldsymbol{k}) = \sin k_1 \Gamma_{01} + \sin k_2 \Gamma_{03} - \left[\frac{1}{2} + \frac{1}{2}(\cos k_1 + \cos k_2) + \frac{3}{2}\cos(k_1 + k_2)\Gamma_{22} + \frac{1}{2}\Gamma_{13}\right]. \quad (I.2)$$

To calculate the $\boldsymbol{r}$-space Euler number in a finite $L \times L$ supercell of the square lattice, we construct the real-space Hamiltonian $H^{(2,2)}$ by performing the Fourier transformation to the Bloch Hamiltonian $H^{\chi_1,\chi_2}(\boldsymbol{k})$, which yields

$$H^{(\chi_1,\chi_2)} = \sum_{ij}\sum_{\mu\nu}\sum_{\boldsymbol{k}\in BZ} e^{i\boldsymbol{k}\cdot(\boldsymbol{r}_i-\boldsymbol{r}_j)}[H^{(\chi_1,\chi_2)}(\boldsymbol{k})]_{\mu\nu}c_{i\mu}^\dagger c_{j\nu}. \quad (I.3)$$

Here, $\boldsymbol{r}_i$ is the lattice vector of the $i$-th site in the square lattice, and $c_{i\mu}^\dagger (c_{i\mu})$ is electron creation (annihilation) operator on the $\mu$ orbital at the $i$-th site. For simplicity, we only consider nearest-neighbor pairs $\langle ij \rangle$ in the lattice. The hopping between site $i$ and $j$ is determined by the summation over $\boldsymbol{k}$ in the BZ, $t_{\mu\nu}(\boldsymbol{r}_{ij}) = \sum_{\boldsymbol{k}\in BZ} e^{i\boldsymbol{k}\cdot(\boldsymbol{r}_i-\boldsymbol{r}_j)}[H^{\chi_1,\chi_2}(\boldsymbol{k})]_{\mu\nu}$. The on-site energies are given by $\epsilon_\mu = t_{\mu\mu}(\boldsymbol{0})$.

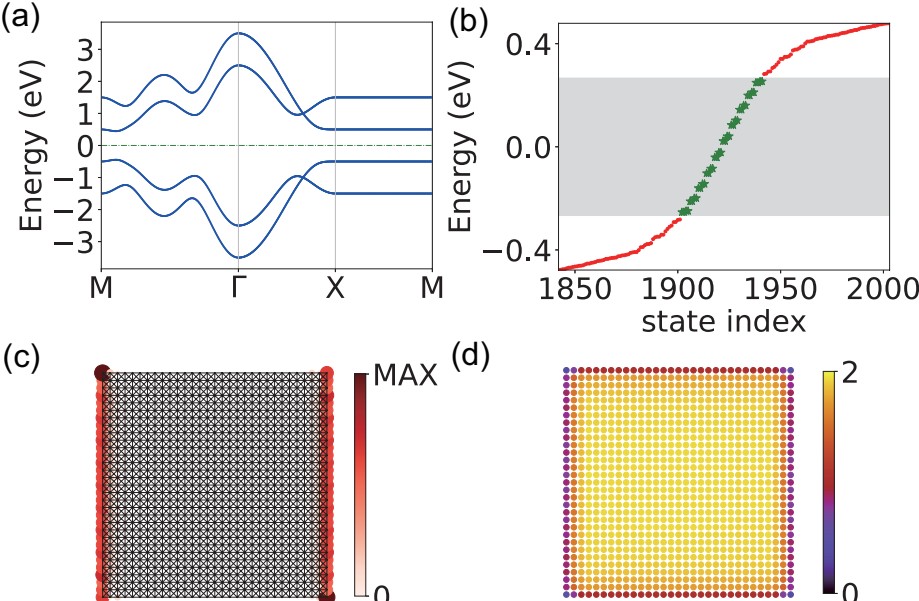

Figure S9: The topological Euler phase with $(\chi_1, \chi_2) = (2, 2)$ in a square lattice based on the minimal four-band model in Eq. (I.3). (a) Band structures of the four-band model in the square lattice. (b) Energy spectrum of a finite sample with OBC. The lattice size is $L = 31$. The bulk gap obtained using PBC is marked in grey. (c) Real-space distribution of the in-gap states [highlighted by green stars in (b)] which are localized on two opposite edges of the finite sample. (d) The real-space distribution of the local Euler marker $e(\boldsymbol{r})$ in the sample with OBC.

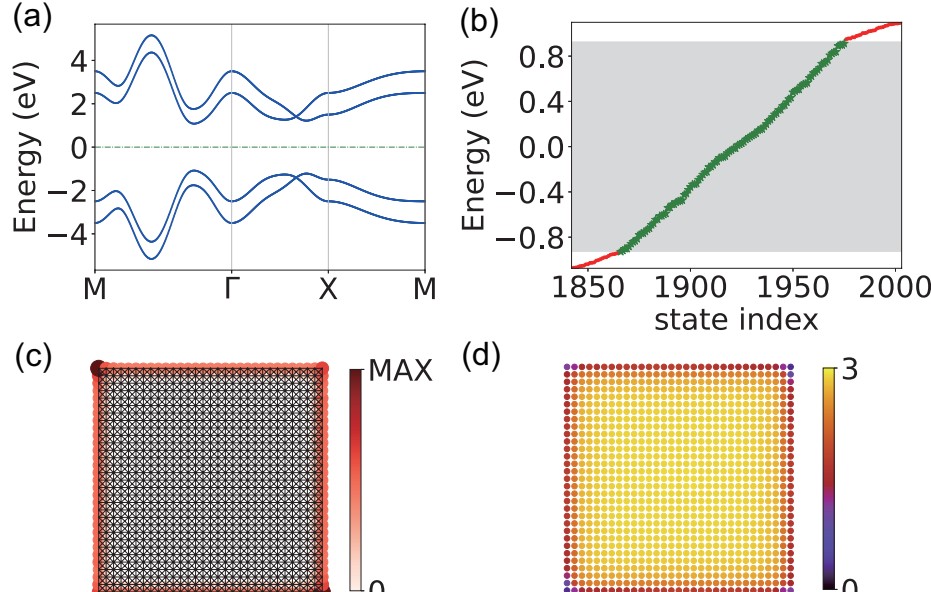

Figure S10: The topological Euler phase with $(\chi_1, \chi_2) = (3, 1)$ in a square lattice based on the minimal four-band model. (a) Band structures of the four-band model in the square lattice. (b) Energy spectrum of a finite sample with OBC. The lattice size is $L = 31$. The bulk gap obtained using PBC is marked in grey. (c) Real-space distribution of the in-gap states [highlighted by green stars in (b)] which are localized on the edges of the finite sample. (d) The real-space distribution of the local Euler marker $e(\mathbf{r})$ in the sample with OBC.

The calculated results are shown in Fig. S9. Similar to the Euler insulator with $e = 1$ presented in the main text, the OBC energy spectrum exhibits some states in the bulk gap. However, these in-gap states are localized on edges instead of corners of the finite sample [see Fig. S9(c)]. This indicates distinct topological behaviors from the topological Euler insulator with $e = 1$. According to the relation between the second Stiefel-Whitney number and the Euler number $w_2 = e \mod 2$, the Euler insulator with $e = 1$ is also a Stiefel-Whitney insulator with $w_2 = 1$ which exhibits higher-order topology with corner states in the presence of additional chiral symmetry [49, 105]. In contrast, the Euler phase with $e = 2$ leads to a trivial second Stiefel-Whitney number $w_2 = 0$. Nevertheless, the system associated with the nonzero Euler number still has a fragile band topology [50]. As shown in Fig. S9 (d), we plot the real-space distribution of the local Euler marker, which exhibits similar bulk domination and edge diminution behavior as those studied in the main text. Remarkably, the local Euler markers inside the bulk are close to the expected value of 2, which results in the averaged $\mathbf{r}$-space Euler number being $e = 2$.

We further validate our $\mathbf{r}$-space Euler number in another four-band model with different Euler numbers for occupied and unoccupied bands. Specifically, we chose the minimal model with $(\chi_1, \chi_2) = (3,1)$, which can be formulated as

$$H^{(3,1)}(\mathbf{k}) = \begin{pmatrix} \bar{a} \\ \bar{b} \\ \bar{c} \end{pmatrix}^{\mathrm{T}} \cdot \Gamma \cdot \begin{pmatrix} \bar{a} \\ \bar{b} \\ \bar{c}' \end{pmatrix}, \tag{I.4}$$

with

$$\Gamma = \begin{pmatrix} -\Gamma_{33} & -\Gamma_{13} & \Gamma_{01} \\ \Gamma_{31} & \Gamma_{11} & \Gamma_{03} \\ \Gamma_{10} & -\Gamma_{30} & -\Gamma_{22} \end{pmatrix}, \tag{I.5}$$

and

$$\bar{a} = \sin k_1 \,,$$
$$\bar{b} = \sin k_2 \,,$$
$$\bar{c} = \frac{1}{2}(1 + (\cos k_1 + \cos k_2) + 3\cos(k_1 + k_2)) \,, \tag{I.6}$$
$$\bar{c}' = \frac{1}{2}(3 - 2(\cos k_1 + \cos k_2) - \cos(k_1 + k_2)) \,.$$

The results of the minimal model with $(\chi_1, \chi_2) = (3, 1)$ are illustrated in Fig. S10. In this case, the unbalanced $|\chi_1| \neq |\chi_2|$ leads to the lack of additional symmetry of the system [106]. Consequently, although the system is a topological phase with nontrivial Stiefel-Whitney number $w_2 = 1$ because of the odd Euler number of the occupied bands, there is no additional symmetry to ensure the localization at the corner. Therefore, this phase does not exhibit the higher-order corner characteristics of conventional Stiefel-Whitney insulators [107–111].

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
