# Peer review of "Real-space approach for the Euler class and fragile topology in quasicrystals and amorphous lattices"

_SciPost Physics, doi:SciPost Phys. 17, 086 (2024)_

## Round 1 · Referee Report · Anonymous (Referee 1) · 2024-6-11

Strengths

  • very interesting subject and questions are addressed
  • written clearly

Weaknesses

  • scientifically it does not solve the question, albeit interesting, that was posed
  • refers to code online that in fact misses the subroutine that would is at heart of solving the issue
  • claims made are not supported or only addressed in limits that were known

Report

The manuscript by Li et al. proposes a real space formalism for describing fragile topology in tight-binding models of topological phases on amorphous lattices and in quasicrystals. In particular, the authors focus on topology captured by the Euler class. An analogue of the Bott index, i.e. a real space indicator known to reflect the Chern topology, is proposed for Euler topology. Furthermore, an attempt to formulate real-space topological Euler markers is provided. The proposals are followed by numerical results for disordered systems, which in clean limit host a fragile topological invariant, supporting the previously obtained results of other groups discovering topological phase transitions in disordered fragile topological systems. Finally, it is argued that the proposed real space indicators can indicate fragile topology in quasicrystals, where the standard symmetries ensuring a reality conditions, such as spatiotemporal inversion, are not present.

The problem of capturing fragile and Euler topology in real space is interesting and timely for the field. As such the questions asked are certainly interesting and highly relevant. However, as argued point-by-point below, one should strongly doubt that the introduced derivation of an appropriate Euler marker in the manuscript is sound. We thus think that the work does not support the claims made by the authors. Several important clarifications and improvements are therefore required. In essence, we note that the definitions only make sense when translational symmetry and proper momentum can be assigned. In contrast, when given a disorder real space system without knowledge of the momentum space Hamiltonian it could adiabatically connect to, the markers are not show/derived to be robust quantities to consider. We detail this in the following.

  • The main criticism is that characteristic classes, such as Euler or Chern, mathematically describe vector bundles. Euler class characterizes real vector bundles of Bloch states, where the base space of the vector bundle is $k$-space; more precisely the BZ torus. Proposing a definition of the $r$-space Euler marker, which by the argumentation provided by the authors appears to be $a~priori$ a definition and is not really a derivation (contrary to the Chern marker for Chern insulators, derived by Bianco and Resta~[Phys. Rev. B 84, 241106(R)]), appears to not reflect the topology of the Bloch vector bundle defining the Euler class as a characteristic class, in general.

  • The reason why this is a severe issue, is that the topological quantization of the invariant stems from the non-triviality of the (k-space) vector bundle of the Bloch states, which, in particular, when non-trivial (corresponding to the non-zero Euler class), manifests its topology by admitting no non-vanishing smooth sections in the Bloch bundle, etc. Within the r-space formulation considered here, there is no rigorous analytical argument for the topological quantization of, or a derivation-based $exact$ correspondence of the proposed Euler marker to, the Euler class invariant in the studied amorphous, disordered, or quasicrystalline systems.

  • Very precisely, starting from the real-space Hamiltonian of a generic inhomogeneous system that satisfies the reality condition (e.g. preserving a C$_2$T symmetry), the authors failed to provide the exact expression of a topological invariant readily expressed in terms of the real-space eigenvectors such that it does not rely on the known clean reciprocal limit from which the Pf$_\text{occ}$ operation is defined (disentangling the “internal space", and defining the “hybridization" mentioned by the authors).

  • This severe issue of the impossibility of evaluation of the Pf$_\text{occ}$ without a reference to the clean bands $\ket{u_{1\textbf{k}}}$, $\ket{u_{2\textbf{k}}}$, appears to be swept under the rug in the code provided by the authors (i.e checking the gitthub).

  • Namely, where the subtle Pfaffian Pf$_\text{occ}$ is expected to be defined, the authors use a mysterious (and most importantly, not included in the code provided!) imported function (“import pf"), which in such state cannot be accepted as a valid and transparent method of computing the Euler marker. Indeed, the code provided only calculates the commutator, while the whole issue resolves around how to define the pfaffian WITHOUT referring to the k space state disentangling.

-As another remark, we note that the general theory that outlines multi-gap and Euler as well as fragile invariants, Physical Review B 102 (11), 115135 (2020), was not cited.

  • finally I would advise to change the title as there is no formalism, neither is there explicit quasiperiodicity studied

Therefore, I cannot recommend the publication of the manuscript in its current form and hence require severe improvements.

Requested changes

see report

Recommendation

Ask for major revision

  • validity: good
  • significance: good
  • originality: high
  • clarity: high
  • formatting: perfect
  • grammar: good

Author:  Huaqing Huang  on 2024-07-22  [id 4642]

(in reply to Report 1 on 2024-06-11)
Category:
answer to question
pointer to related literature

We thank the Referee very much for his/her insightful reading of the paper and extremely constructive report. We have made some revisions and added some references following his/her suggestions. We believe it has improved the quality and clarity of the paper. Below we answer point by point and describe the changes made.

Comment:
“-The main criticism is that characteristic classes, such as Euler or Chern, mathematically describe vector bundles. Euler class characterizes real vector bundles of Bloch states, where the base space of the vector bundle is k-space; more precisely the BZ torus. Proposing a definition of the r-space Euler marker, which by the argumentation provided by the authors appears to be a priori a definition and is not really a derivation (contrary to the Chern marker for Chern insulators, derived by Bianco and Resta~[Phys. Rev. B 84, 241106(R)]), appears to not reflect the topology of the Bloch vector bundle defining the Euler class as a characteristic class, in general. ”

Response:
We appreciate the referee's insightful comments. It is important to note that the Chern marker, similar to the Euler marker we propose, is essentially a definition rather than a derivation. In Bianco and Resta's original paper [Phys. Rev. B 84, 241106(R)], the pivotal steps leading to the local marker are found in Eq. (3) and Eq. (9), which facilitate the transition from k-space to real space and from a global quantity to a local quantity, respectively. They explicitly stated that these equations “should be interpreted as definitions” of the corresponding quantities in the main text just below these equations. While their paper does not provide a rigorous mathematical foundation for the concept of a local Chern marker, the numerical results presented were compelling and have had a significant impact on the field. In this context, we believe that our manuscript similarly introduces a valuable new local marker, which provides a possible local description of topological order due to the “nearsightedness” of the ground-state density matrix, just as emphasized by Bianco and Resta in their concluding remarks.

Comment:
“-The reason why this is a severe issue, is that the topological quantization of the invariant stems from the non-triviality of the (k-space) vector bundle of the Bloch states, which, in particular, when non-trivial (corresponding to the non-zero Euler class), manifests its topology by admitting no non-vanishing smooth sections in the Bloch bundle, etc. Within the r-space formulation considered here, there is no rigorous analytical argument for the topological quantization of, or a derivation-based exact correspondence of the proposed Euler marker to, the Euler class invariant in the studied amorphous, disordered, or quasicrystalline systems.”

Response:
We appreciate the referee's detailed feedback. The theory of characteristic classes of vector bundles is indeed foundational for topology defined in k-space. However, this k-space formalism becomes inadequate when dealing with real space, especially in the presence of certain types of disorder. In such cases, non-commutative geometry, as introduced by J. Bellissard et al. [J. Math. Phys. 35, 5373–5451], serves as a robust mathematical framework. In their work, they defined the non-commutative Brillouin zone (BZ) to include the effects of disorder, thereby allowing topological quantities in k-space to be generalized to real-space equivalents.

Non-commutative geometry is thus well-suited for real-space topological invariants, such as the Chern number and Bott index, as demonstrated in the literature [Lett. Math. Phys. 112, 126]. It admits that the real-space expression may not provide an exact quantized value, however, the real-space formula is well-defined on a finite torus and coincides up to corrections of order $O(L^{-1})$ with the integer. Our definition of the local Euler marker is inspired by the discussions in sections III.A and III.B of the pioneering work of Bellissard, van Elst, and Schulz-Baldes [J. Math. Phys. 35, 5373–5451]. While our manuscript may not attain the same level of mathematical rigor as the referenced work [J. Math. Phys. 35, 5373–5451], we are confident it offers valuable insights and perspectives that will be of interest to both physicists and mathematicians interested in this field.

Accordingly, we have revised our manuscript to include the following statement:
“To generalize a formula of topological system defined in k space to its real-space form applicable to disordered system, a standard mathematical framework is the non-commutative geometry, which provides the duality (see the equivalence at least for translational invariant systems in Appendix C)”

Comment:
“-Very precisely, starting from the real-space Hamiltonian of a generic inhomogeneous system that satisfies the reality condition (e.g. preserving a C22T symmetry), the authors failed to provide the exact expression of a topological invariant readily expressed in terms of the real-space eigenvectors such that it does not rely on the known clean reciprocal limit from which the Pfocc operation is defined (disentangling the “internal space", and defining the “hybridization" mentioned by the authors).
-This severe issue of the impossibility of evaluation of the Pfocc without a reference to the clean bands \ketu1k, \ketu2k, appears to be swept under the rug in the code provided by the authors (i.e checking the gitthub).
-Namely, where the subtle Pfaffian Pfocc is expected to be defined, the authors use a mysterious (and most importantly, not included in the code provided!) imported function (“import pf"), which in such state cannot be accepted as a valid and transparent method of computing the Euler marker. Indeed, the code provided only calculates the commutator, while the whole issue resolves around how to define the pfaffian WITHOUT referring to the k space state disentangling.”

Response:
We appreciate the referee's detailed comments.

First, we want to clarify that the function "pf" in our code is simply a standard Python function used to numerically calculate the Pfaffian. It can be understood as extracting the off-diagonal element when $N_{occ}=2$, and it is unrelated to the construction of Wannier functions.

Given the potential confusion surrounding the use of Wannier functions, we will clarify the conceptual framework of our manuscript. Our work proposes a new local Euler marker, the average of which can be interpreted as the real-space Euler number according to the standard non-commutative geometry paradigm. However, this marker can only be calculated on the Wannier basis, necessitating the construction of such a basis in systems lacking translational invariance.

A straightforward approach to constructing real-space Wannier functions is through functional optimization, where the initial choice of the Wannier function is critical. In our case, this initial value can, in principle, be set arbitrarily. Nevertheless, for improved efficiency and convergence, the initial function is often chosen as the Wannier function derived from a translationally invariant system. Specifically, for disordered lattices, the initial Wannier function constructed through the Fourier transformation of Bloch functions from a perfect lattice serves well. This approach, however, becomes challenging for quasicrystals and amorphous systems. In our work, we derived the initial Wannier function for quasicrystals from a 16×73 rectangular lattice (as depicted in Fig. 2).

For fully non-periodic systems where gauge optimization is ineffective, other methods, such as the Iterated Projected Position (IPP) algorithm, should be considered as they do not require initialization [Phys. Rev. B 103, 075125].

Accordingly, we have made the following revisions to our manuscript:
“Since the real-space Euler number obtained in Eq. (6) can only be calculated on Wannier basis, a crucial step in the numerical calculation is to construct the Wannier function in systems even without the spatial translational symmetry.
One possible way to construct the real-space Wannier function is the functional optimization method.
In the functional optimization process, a key factor is the selection of the initialization. In our case, this is the initial value of Wn. In principle, the initial Wannier function can be set arbitrarily. However, to obtain a more efficient and convergent result, the initial function can be chosen as the Wannier function obtained in a translational invariant system. To be more specific, for disordered lattices, the Wannier function constructed through the Fourier transformation of the Block functions obtained in perfect lattice is a great initial function. However, it might be hard to find such a k-space analog in quasicrystal and even totally amorphous systems. In our work, the initial Wannier function of the quasicrystal is obtained from that of a 16×73 rectangle lattice in Fig. 2.
As for the fully non-periodic systems where the gauge optimization fails to work, other methods such as the Iterated projected position (IPP) algorithm are supposed to be considered without the initialization requirement [104].”

Comment:
“-As another remark, we note that the general theory that outlines multi-gap and Euler as well as fragile invariants, Physical Review B 102 (11), 115135 (2020), was not cited.”

Response:
We appreciate the referee's suggestion. We have now cited this paper in our manuscript to acknowledge the contributions of relevant prior research.

Comment:
“-finally I would advise to change the title as there is no formalism, neither is there explicit quasiperiodicity studied”

Response:
We thank the referee for this suggestion. In the manuscript, we have provided a real-space approach for the Euler class and applied it to explore the Euler topology in quasicrystal and amorphous lattices. According to the referee’s suggestion, we have replaced the formalism with an approach in the title.

---

## Round 1 · Referee Report · Anonymous (Referee 2) · 2024-6-16

Strengths

  1. The authors have proposed a real-space Euler marker to characterize the topology of the Euler class. Such a formalism is very important as it can be used to identify the Euler topology in non-crystalline systems, such as quasicrystals and amorphous systems.

  2. The paper is well written and organized.

Weaknesses

  1. The authors do not specify what |r> denotes in Eq. (7). This state is actually the Wannier function. However, when I first read it, I thought that it would the position state. Hope that the authors can provide the definition immediately following Eq. (7) for a better readability.

  2. I wonder whether the submatrix defined over the Wannier basis in r-space is still antisymmetric in the absence of spatial translational symmetry. We know that the Pfaffian is defined only for antisymmetric matrices. In addition, how is the order of |W_1(R)> and |W_2(R)> determined in the calculations? This is important as the signs of the Pfaffian for distinct order are opposite.

  3. I think Fig. S3 and Fig. 1(d) should use logarithmic coordinates to better illustrate the power-law decay of (1-|e|) versus L.

Report

In the manuscript, the authors propose a real-space formalism of the topological Euler class by constructing a local Euler marker. Based on such a novel marker, the Euler topology in disordered systems, quasicrystals and amorphous systems is identified. In particular, the Euler topology supporting eight corner modes is found in a quasicrystal. I thus think that the results are very interesting and important, making a significant advancement in the field. I thus can recommend the paper for publication in SciPost Physics after some minor revisions.

Requested changes

  1. Please provide the specific relation between $t_{\mu\nu}$ and ($V_{dd\sigma}$, V_{pp\pi}, $\dots$).
  2. Please provide the energy spectrum under PBCs and OBCs, and the spatial distribution of nontrivial edge states for a specific $\Delta$ in region I of Fig. 1(b). The information can be compared with Fig. 3 to illustrate the upward shift of eigenenergies of corner states.
  3. Please correct the citation “appendix ??” in line 188.
  4. $a_E/1-|e|$” should be written as $a_E/(q-|e|)$ in line 608 and Fig. S2.
  5. The disorder parameter of the amorphous lattice in Fig. 3 is missing. Please add the information.
  6. The construction methods of the amorphous lattice in Fig. S6(a) as mentioned in line 539 and line 661 are completely different. Please double check it.

Recommendation

Ask for minor revision

  • validity: top
  • significance: top
  • originality: high
  • clarity: high
  • formatting: excellent
  • grammar: excellent

Author:  Huaqing Huang  on 2024-07-22  [id 4641]

(in reply to Report 2 on 2024-06-16)
Category:
answer to question

We would like to thank the referee for his/her high assessment of our work. We respond to the comments below and make corresponding changes in the revised manuscript.

Comment 1: “The authors do not specify what |r> denotes in Eq. (7). This state is actually the Wannier function. However, when I first read it, I thought that it would the position state. Hope that the authors can provide the definition immediately following Eq. (7) for a better readability.”

Response 1: We thank the referee for reading our manuscript carefully. Accordingly, we have made the following revisions in our MS:
“where |r> denotes the basis to construct the external space indexed by the Wannier cell r.”

Comment 2: “I wonder whether the submatrix defined over the Wannier basis in r-space is still antisymmetric in the absence of spatial translational symmetry. We know that the Pfaffian is defined only for antisymmetric matrices.”

Response 2: We thank the referee for this important question. The anti-symmetry required for the definition of Pfaffian is indeed ensured by the structure of the expression itself. To clarify, we present the submatrix (denoted as A here in both k and r space) in its original form as shown in Eq. (2) in the main text, which can be written as $\langle\partial_{[k_x}u_1(k)|\partial_{k_y]}u_2(k)\rangle$. Due to the commutator, an additional sign appears from the interchange of 1 and 2, ensuring the anti-symmetry of the submatrix. Therefore, considering $A_{12}$, one off-diagonal element, effectively defines the Pfaffian.

This discussion extends to Eq. (6), where the submatrix $A=\hat{P}[\hat{U}\hat{P}\hat{U}^\dagger,\hat{V}\hat{P}\hat{V}^\dagger]$ in r space also involves a commutator. Specifically, $A_{mn}=A^*_{mn}=A^\dagger_{nm}=-A_{nm}$, implying $A=-A^T$. Therefore, the submatrix inherently preserves anti-symmetry even in the absence of spatial translational symmetry.

Comment 3: “In addition, how is the order of |W_1(R)> and |W_2(R)> determined in the calculations? This is important as the signs of the Pfaffian for distinct order are opposite.”

Response 3: We thank the referee for this important question. Analogous to the k-space case, the order of the Wannier basis can be generally determined by the diagonal elements of the Hamiltonian in this basis. Additionally, any sign change will manifest in the distribution of the local Euler markers. Hence, the order can also be determined by ensuring the continuity of the local Euler markers, thereby maintaining consistency throughout the calculations.
Accordingly, we have added a paragraph at the last of Appendix H.5 in our MS:
“Another issue to be clarified is the ordering of occupied states within a certain cell r. It can be determined by the corresponding diagonal element of the Hamiltonian on the composite Wannier basis. We also noticed that the local Euler marker is attached to only an additional minus sign when this ordering is inverted. Therefore, more conveniently, the sign of local Euler markers can be set to satisfy the continuity of these markers.”

Comment 4: “I think Fig. S3 and Fig. 1(d) should use logarithmic coordinates to better illustrate the power-law decay of (1-|e|) versus L.”

Response 4: We thank the referee for this helpful suggestion. Accordingly, we have updated Fig. S3 and Fig. 1(d) in our MS to use logarithmic coordinates.

Comment 5: “Please provide the specific relation between $t_{\mu\nu}$ and ($V_{dd\sigma}$, $V_{pp\pi}$, ……).”

Response 5: We thank the referee for this suggestion. Accordingly, we have added all these specific relations in Appendix H.1 in our MS.

Comment 6: “Please provide the energy spectrum under PBCs and OBCs, and the spatial distribution of nontrivial edge states for a specific $\Delta$ in region I of Fig. 1(b). The information can be compared with Fig. 3 to illustrate the upward shift of eigenenergies of corner states.”

Response 6: We thank the referee for this suggestion. Accordingly, we have added a new section with a new Figure S8 in Appendix I.7 of the MS:
“In this section, we discuss the upward shift of eigenenergies of corner states with increasing on-site energy. As illustrated in Fig. 3 and Fig. S7, introducing structural disorder leads to the upward shift of the eigenenergies of corner states. In fact, this effect originates from the decreasing energy gap. In structurally disordered samples, the decrease is attributed to the increasing disorder amplitude. Additionally, the adjustment of the on-site energy can also lead to a smaller bulk gap. As discussed in Appendix I.1, in region I, we can lift the on-site energy of p-orbitals, causing the bulk gap to decrease and eventually vanish at critical point $\Delta_1$. Therefore, for comparison purposes, we consider a square model with the parameters the same as in Fig. 3 except for the on-site energy difference. As illustrated in Fig. S8(a), increasing the on-site energy difference shows a similar upward shift effect to that observed in structurally disordered lattices. These shifted states near the upper bound of the PBC gap are spatially localized at four corners, as shown in Fig. S8(b). These results show the similarity between the effect of on-site energy difference and structural disorder on the upward shifting, which can be explained as the effect of the decreasing bulk gap.”

Comment 7: “Please correct the citation “appendix ??” in line 188.”

Response 7: We thank the referee for carefully reading our manuscript and pointing out this mistake. Accordingly, we have corrected this citation and added a new section with Figure S5 in Appendix I.4 of the MS:
“In this section, we discuss the deviation of r-space Euler number with OBC. The OBC case shows a similar linear dependence between 1/L and the numerical deviation $\Delta e=1-|e|$ with slower convergent behavior. This means that the OBC includes an additional effect which is up to order $O(1/L)$ as well. Notice that the Euler number is obtained by averaging the local Euler markers at all sites. Since the sites far from boundaries are supposed to preserve similar properties to those in periodic systems, such deviation originates from the sites close to the boundary, which contributes $O(L_{edge}/A)=O((L^2-(L-2)^2)/L^2)=O(4L/L^2)=O(4/L)$ as expected. Here $L_{edge}$ and $A$ are the number of sites in the boundary and the whole sample, respectively. This additional factor accounts for the slope approximated to 1/4 in Fig. S5, confirming the effect on r-space Euler number from the edge.”

Comment 8 : “$a_E/1−|e|$ should be written as $a_E/(q−|e|)$ in line 608 and Fig. S2.”

Response 8: We thank the referee for this suggestion. Accordingly, we have made the necessary revisions in the MS.

Comment 9: “The disorder parameter of the amorphous lattice in Fig. 3 is missing. Please add the information. The construction methods of the amorphous lattice in Fig. S6(a) as mentioned in line 539 and line 661 are completely different. Please double check it.”

Response 9: We thank the referee for carefully reading our manuscript and pointing out these issues. The methods to introduce structural disorder in cases in Fig. 3 and original Fig. S6 (Now Fig. S7 in the revised manuscript) are different. In Fig. 2, we study a finite amorphous lattice constructed by assigning random site displacements away from their equilibrium position in an initial square lattice. Consequently, all spatial symmetries are broken. However, in Fig. S7, we specifically preserve the inversion symmetry, for comparison with the case breaking all spatial symmetries presented in Fig. 3 of the main text. As a result, we find that the real-space method works for amorphous lattices without and with inversion symmetry.
Accordingly, we have added the following information to the caption of Fig.3:
“In the amorphous lattice, each atom is assigned with a random displacement following the Gaussian distribution with standard deviation $\sigma = 0.2$.”
We also made the following revisions in Appendix H.3 of the MS to accurately describe the methods used:
“The amplitude d of atomic displacements are determined by Gaussian distributions with standard deviation $\sigma = 0.2$.”

---

## Round 1 · Referee Report · Anonymous (Referee 3) · 2024-7-11

Strengths

  1. The question addressed in the manuscript: develop a local topological marker to describe “fragile” topological systems is both timely and interesting.

  2. The clarity of exposition is of a good standard.

  3. The authors provide a number of applications (and checks) of the proposed formalism. In particular, the authors apply it to a quasicrystal where the conventional method (computing Euler invariant in k space) is not directly applicable due to lack of translational invariance.

Weaknesses

  1. The manuscript in my opinion is lacking a sufficiently careful and clear development of the formalism.

Report

In this manuscript, a real-space invariant to diagnose fragile topological states is proposed. The derived expression is used to investigate amorphous and quasicrystalline systems, both lacking translational invariance.

I think the topic is a great idea! I think the work should be published. I would like to provide some feedback, however. This is in the next field.

Requested changes

1.
The operator involved entering (for instance) equation (6) is P[UPU^dagger, VPV^dagger]. This operator also enters the Chern marker. One can see that it is directly related to the field strength matrix when there is translational invariance. So to me it seems to be a fairly direct *observation* — to obtain the Euler number you just need to replace Trace —> Pfaffian.

I think the readability might be improved if this is emphasized. It is fairly direct.

2.

What’s less clear to me is how the required symmetries translate to the real space expressions. As I understand, the integrated expression curl A_{12} is topological only when the u(k) can be taken to be real. Indeed, this makes the field strength matrix skew symmetric which is needed to even define the Pfaffian.

Please clarify:
How does this all translate to the real space expressions?
What are the required symmetries of the Hamiltonians considered and what does this imply about P[UPU^dagger, VPV^dagger]?

Recommendation

Publish (easily meets expectations and criteria for this Journal; among top 50%)

  • validity: high
  • significance: high
  • originality: high
  • clarity: ok
  • formatting: excellent
  • grammar: excellent

Author:  Huaqing Huang  on 2024-07-22  [id 4640]

(in reply to Report 3 on 2024-07-11)
Category:
answer to question

We were happy to read the Referee's positive report and thank the Referee for reviewing our manuscript. We address the two comments below:

Comment 1: “The operator involved entering (for instance) equation (6) is P[UPU^dagger, VPV^dagger]. This operator also enters the Chern marker. One can see that it is directly related to the field strength matrix when there is translational invariance. So to me it seems to be a fairly direct *observation* — to obtain the Euler number you just need to replace Trace —> Pfaffian.
I think the readability might be improved if this is emphasized. It is fairly direct.”

Response 1: We thank the referee for this suggestion. Accordingly, we have added a remark in our MS:
“Formally, Eq. (6) share a similar expression to the real-space Chern number except for the substituting from Tr to Pfocc.”

Comment 2: “What’s less clear to me is how the required symmetries translate to the real space expressions. As I understand, the integrated expression curl A_{12} is topological only when the u(k) can be taken to be real. Indeed, this makes the field strength matrix skew symmetric which is needed to even define the Pfaffian.
Please clarify:
How does this all translate to the real space expressions?
What are the required symmetries of the Hamiltonians considered and what does this imply about P[UPU^dagger, VPV^dagger]?”

Response 2: We thank the referee for this important question.

(1). Translation to Real-Space Expressions:
The reality condition of u(k) is crucial for the Euler number, as discussed in previous literatures (For example, Section C of the supplementary material of [Nat. Phys. 16, 1137–1143]). For any spinless time-reversal invariant model, the time-reversal operator T satisfies: (i) $T^2=1$ and (ii) $THT^{-1}=H$.
After the Takagi decomposition, we can always find a basis in which the Hamiltonian is a real symmetric matrix. A real symmetric matrix is orthogonally diagonalizable, meaning its eigenfunctions can also be taken to be real.

(2). Required Symmetries and Implications:
The antisymmetry of the operator $P[UPU^\dagger, VPV^\dagger]$ is ensured by the reality condition and the expression itself. Specifically, for the submatrix A: $A_{mn}$=$A_{mn}^*$=$A^\dagger_{nm}=-A_{nm}$. This implies $A=-A^T$, maintaining antisymmetry even in the absence of spatial translational symmetry.
To summarize, the reality condition ensures that the eigenfunctions $u(k)$ can be taken as real, leading to a skew-symmetric field strength matrix. This antisymmetry is preserved in the real-space formulation through the structure of the operator $P[UPU^\dagger, VPV^\dagger]$, thus allowing the definition of the Pfaffian and ensuring the topological nature of the Euler number in real space.

---

## Round 2 · Referee Report · Anonymous (Referee 2) · 2024-7-30

Report

I thank the authors for carefully addressing my questions and suggestions. In the revised manuscript, the authors have added the discussion on composite Wannier functions and the corresponding ordering, and improved the presentation of the results. Since the results are very interesting and important, thus making a significant advancement in the field, I can recommend the manuscript for publication in SciPost Physics.

Recommendation

Publish (meets expectations and criteria for this Journal)

  • validity: top
  • significance: high
  • originality: high
  • clarity: high
  • formatting: excellent
  • grammar: excellent

Author:  Huaqing Huang  on 2024-08-19  [id 4703]

(in reply to Report 1 on 2024-07-30)

We sincerely thank the referee for the thoughtful review and for recognizing the significance of our work. We are glad that the revisions and additional discussions we provided have addressed the questions and suggestions by the referee.

---

## Round 2 · Referee Report · Anonymous (Referee 1) · 2024-8-5

Strengths

  • Interesting topic
  • Interesting and timely question

Weaknesses

  • No proof of actual method, rather vague references to fundamental instead
  • Response to fundamental questions at the heart of the method
  • No actual computational code [incomplete and empty, missing main part] or description of most computationally demanding details

Report

I thank the authors for their answers. Regrettably, I still cannot recommend the acceptance of this method as a valid general real-space approach to Euler topology, on which I further elaborate below. As I am very well informed in this subject, I am a bit disappointed by some of the answers and especially the fact that some of my insights were twisted.

A case in point is the *whole* construction, i.e. the core of the paper (even at the analytical level), and a relation to the Pfaffian invariant. As I pointed out repeatedly, while the question is interesting, the question is hard in the sense that the Euler invariant is a multi-gap invariant, as originally defined in the momentum space formalism. There, on taking a patch $\mathcal{D}$ in the BZ the invariant reads $\chi = \frac{1}{2\pi} \int_{\mathcal{D} \in \text{BZ}} d ^{2} \textbf{k}~\text{Eu} - \frac{1}{2\pi} \oint_{\partial\mathcal{D}} d \textbf{k} \cdot \vec{a}$, where $\vec{a}$ is the Euler connection defined in terms of the Pfaffian of the non-Abelian Berry connection $\vec{A}_{n,n+1}(\mathbf{k}) = \langle u_n|\nabla_{\mathbf{k}} u_{n+1}\rangle $ and $\text{Eu} = \nabla_{\textbf{k}} \times \vec{a}$ is the Euler curvature [Nature Physics vol. 17, p. 1239–1246 (2021)]. So crucially the Euler form takes/combines/compares *different* states, by taking an inner product between two band states that form the two-band subspace. Now the crucial question is how is this multi-band/multi-state form defined in real space WITHOUT translational symmetry. Only when connecting to that limit the discussion makes sense, but this is not truly a marker as in the Chern case [\textit{Note that here this problem does not arise the same way. For the Chern invariant, one considers the projector-reformulated curvature for individual states on comparing with themselves (hence arises the trace structure), i.e the diagonal components of position/derivative operators between the same states with the same band indices are considered, as the Chern invariant involves projectors onto single/individual bands, unlike the multi-band Euler invariant}]. Indeed if one does not know the bands [requiring translational symmetry] how does one disentangle the state states in the real-space Pfaffian? For a true (general/universal) marker one is given the disordered system and should evaluate THIS system to see non-trivially, as is done for the Chern marker without the knowledge of the states in translationally invariant case.

It was precisely in view of the above point that I checked how this is done in the code. I am surprised that the authors asserted that I did not check whether "pf" is a "standard Python module" or not, before providing my criticism. The list of "standard Python modules" is available here: https://docs.python.org/3/py-modindex.html. Alas, the imported "pf" is not listed amongst the index of "standard Python modules", which makes the method questionable to me, to say the least. Then one notices that apart from a simple commutator the code calls a function "pf" that is not defined, one could try to "pip install pf", in case it was not a standard module, as the authors claimed. The installed module (https://pypi.org/project/pf/) turns out to be the "Project Files templating engine"... Now, "pf" can be called indeed. However, with the importation implemented by the authors, it calls not for a Pfaffian, but for the "project file" (standing for "pf") management.

Apart from the ironic fact that there seems to be no standard "pf" function in Python, my question revolved around how one defines the Euler invariant [separating the states if there is no translational symmetry]. While anyone can take a Pfaffian of an antisymmetric matrix (especially in momentum-space Bloch state basis, with ``$N_{occ} = 2$", as the authors mention), the whole point is that the authors claim they can compute a marker in absence of well-defined momentum states (I am not even convinced if in a real-space basis, the operator on which the Pfaffian is supposed to operate is generally antisymmetric in the real-space basis within the proposed approach), which is needed to compute the invariant in a concise manner, refering only to the real-space eigenstates. This is completely missing as a whole in the code, which apart from simple computing a (standard) commutator (as for the Chern markers...) gives no insights.

Moreover, it should be stressed that the computation of Pfaffians in and extended, such as real-space, basis (for a real-space Euler invariant, rather than the marker) is numerically a *very* costly task [ACM Trans. Math. Software 38, 30 (2012)]. Namely, rather than computing a "square root of a determinant" for a $2 \times 2$ matrix (in momentum-space); here, one needs to compute a "square root of a determinant" for, say, a $2000 \times 2000$ matrix, to get the Pfaffian. I am not quite sure how the unspecified "standard Python module" can handle that serious task.

Similar issues arise with their answer referring to the work of Bellissard. Again, I am familiar with this high end work, as well as the work of Connes. The point is that using the notions of non-commutative Brillouin zones generalizations of Chern numbers [again single bands] are introduced. It is proven that they are invariants that doesn't mean that without any further insight one can repeat this for the multi-gap Euler case. The whole point is that this has to be shown to be an adequate setup and producing a marker as claimed.

Also in view of the reference to the work of Bianco and Rsta I must note an important difference. First as there they consider the single Chern band case one can effectively compute the commutator and show these sum (with a trace, rather than a Pfaffian) to the Chern number. It is furthermore also well established in the context of Thouless pumping and the modern theory of polarization how these concepts hang together (based on the trace structure). Again the multi-gap nature in which one has a trace over a Pfaffian over two states, which involves mathematical operations that do not commute, unlike a combination of two traces, makes it even more complicated. The question of how to define this in absence of well defined bands as function of momentum is an aspect that is the core of the problem and not answered at all by the authors. They just claim this can be done, although not clear how, other than referring to very general statements like the ones above, incl. the use of "standard Python modules" to compute the troublesome Pfaffian.

Finally, I stress that the authors should note the role of multi-gap topology. Fragile topology means invariants beyond K-theory that can be trivialized by adding trivial bands, contrary to stable invariants such as Chern numbers or $Z_2$ QSHE invariants that need a gap closing with bands having opposite invariants. However fragile topologies can [and indeed the first paper by Po, Watanabe, and Vishwanath is an example] can be symmetry indicated. In that case they are simply the difference between two elementary band representations that represent an atomic limit. The Euler class is arising in a two-band subspace due to the partitioning of bands [Physical Review B 102, 115135 (2020)] and the value can be seen as being changed due to the brading of band nodes between different gaps [Nature Physics 16, 1137-1143 (2020) and Physical Review Letters 125, 053601 (2020)] this can be completely symmetry indicator-free-i.e for bands with all same irreps (as in the case of trivial bands). This is not really addressed by the presented real-space approach, and is not reflected in the paper.

All in all, given the other reports, I anticipate that this paper will be (and probably can be) published, but I think that these reservations should at least be mentioned and acknowledged, rather than being swept under the rug with references to general well established theories that do not a priori apply here. Indeed, I find it rather obscure of how the authors retrieve such impressive numerics with the non-existing "standard Python module" applied within a (put-mildly) questionable real-space approach, to such a complicated problem.

Requested changes

see report; main issue is to fix or at (*least acknowledge*) the problem with defining the Pfaffian marker in real space without referring to translationally invariant case.

Recommendation

Ask for minor revision

  • validity: poor
  • significance: good
  • originality: good
  • clarity: good
  • formatting: perfect
  • grammar: good

Author:  Huaqing Huang  on 2024-08-19  [id 4701]

(in reply to Report 2 on 2024-08-05)

We sincerely appreciate the referee’s time and effort in reviewing our manuscript. We are also grateful for the insightful remarks, which indeed highlight some important shortcomings in our work.

We understand the referee’s concerns regarding the definition of the Euler invariant in real space without translational symmetry and the challenges associated with the Pfaffian computation. The key point here is the adaptation of the Euler invariant to a real-space framework, which, as the referee points out, differs significantly from the momentum-space formalism where the invariant is traditionally defined. We acknowledge that the construction of the Euler invariant in real space is indeed a complex task. Our approach relies on mapping the problem into a framework where the Pfaffian can still be meaningfully computed despite the lack of translational symmetry. We will revise the manuscript to highlight the limitations and challenges of this approach.

Furthermore, we recognize that the reliance on standard computational tools and modules requires greater transparency. We regret any confusion caused by the reference to a "standard Python module" for computing the Pfaffian. And we have uploaded the missing python module now. This python module is only used to compute the pfaffian of 2×2 sub-matrices. The referee correctly points out the significant numerical complexity involved in computing the Pfaffian for large matrices in a real-space basis. The computation of Pfaffians in such extended bases is indeed a computationally demanding task. The separation concerned by the referee is established by the construction of the Composite Wannier function, which is indeed numerically time-consuming. Therefore, we think future efforts are required to develop new algorithms with improved convergent rates that can be effectively applied to fully amorphous lattices.

Additionally, due to the multi-gap nature of the Euler number, it is necessary to construct the real-space Wannier functions that separate the internal space from the whole space during numerical calculations. Although several examples are provided in the main text, a purely amorphous case without reference to the translational invariant lattice is still absent. Furthermore, a rigorous mathematical framework has not yet been established. Future efforts are expected to focus on applying non-commutative geometry tools to the case of the Euler number. We hope that our work can inspire future research on developing real-space approach within the broader context of non-commutative geometry, while also clarifying the connection to the well-established momentum-space formalisms. This includes a more rigorous discussion of how the multi-gap nature of the Euler invariant complicates its real-space computation and how to overcome these challenges.

In summary, we fully acknowledge the limitations of our current approach and the need for further theoretical development to rigorously establish a real-space Euler marker. In light of the referee’s comments, we have included a paragraph in our revised manuscript that explicitly acknowledges these limitations and outlines potential future directions for research, including the application of non-commutative geometry tools to fully address the complexities associated with the Euler invariant.

Anonymous on 2024-08-19  [id 4704]

(in reply to Huaqing Huang on 2024-08-19 [id 4701])

I thank the authors for acknowledging my points of the previous reports. Given that it is what it is it, and more importantly that this is now marked, I can endorse publication.

---

## Round 2 · Referee Report · Anonymous (Referee 3) · 2024-8-6

Report

I thank the authors for addressing my points. I have also read carefully the correspondence with Referee 2 who raises very valid points and has gone deeply into details and has valid points.

I am comfortable with recommending publication at this stage. The authors have proposed and motivated a topological marker and demonstrated its utility in a number of contexts. Future work may put the marker on a firmer theoretical foundation.

Recommendation

Publish (easily meets expectations and criteria for this Journal; among top 50%)

  • validity: -
  • significance: -
  • originality: -
  • clarity: -
  • formatting: -
  • grammar: -

Author:  Huaqing Huang  on 2024-08-19  [id 4702]

(in reply to Report 3 on 2024-08-06)

We sincerely appreciate the referee for his/her thoughtful review and positive recommendation for our manuscript. We are pleased that you find our work suitable for publication and we agree that future research will be important for further strengthening the theoretical foundation of the proposed topological marker.

---

## Round 2 · Author Response

Dear Editor,

Thank you for handling our manuscript, and we thank the referees for their helpful comments, which we found useful and constructive, and have helped us to further clarify several technical points in the paper. We have responded to all the comments and suggestions by the referees, and accordingly revised the manuscript. Attached below please find our revised manuscript, responses to all comments and suggestions by the referee, and a summary of the changes made. We trust that the revised manuscript can now be accepted for publication in SciPost Physics.

Sincerely,

Huaqing Huang

/On behalf of all Authors/

---

## Round 2 · List of Changes

According to the referees’ suggestions, we have made the following revisions (Note: all the line numbers mentioned in the following revision list are given according to the revised manuscript):

  1. Added a citation in line 60.
  2. Added the following paragraph “To generalize a formula ... in appendix C” in line 96.
  3. Added the following sentence “Formally, Eq. (6) share a similar expression to the real-space Chern number except for the substituting from Tr to Pfocc.” in line 110.
  4. Added the following sentence “where |r> denotes the basis to construct the external space indexed by the Wannier cell r.” in line 114.
  5. Added the following sentence “(See details in appendix H.1)” in line 152.
  6. Revised the following sentence “but this difference can be diminished ...(see appendix ??)” to “but this difference can be diminished by increasing lattice size (see appendix I.4)” in line 194.
  7. Added the following sentence “Each atom is assigned with a random displacement following the Gaussian distribution with standard deviation σ = 0.2.” in the caption of Fig. 3.
  8. We used another distinct fraktur font “r” to denote all the Wannier cell of the specific basis |r>.
  9. Added the following paragraph “To be more specific, the explicit expression of Slater-Koster ... is the unit direction vector.” in line 519.
  10. Added the following sentence “The amplitude d of atomic displacements are determined by Gaussian distributions with standard deviation σ = 0.2.” in line 549.
  11. Added the following paragraphs “Since the real-space Euler number ... functional optimization method. ” in line 571.
  12. Added the following paragraphs “In such functional optimization process ... without the initialization requirement.” in line 584.
  13. Added the following paragraph “Another issue to be clarified... the continuity of these markers.” in line 596.
  14. Added the following sentences “q-|e| with ... in this case” in the caption of Fig S2.
  15. Added the figure S5 and the section I.4 in line 661.
  16. Added the figure S8 and the section I.7 in line 717.
  17. Revised Figure 1 and Figure S3.

---

## Round 3 · Author Response

Dear Editor,

Thank you for handling our manuscript, and we also would like to express our sincere gratitude to the referees for their careful review of our manuscript and their constructive comments. We have responded to the comments by the referee, and accordingly revised the manuscript. We hope that the revised manuscript can now be accepted for publication in SciPost Physics.

Sincerely,

Huaqing Huang

/On behalf of all Authors/

---

## Round 3 · List of Changes

According to the referees’ suggestions, we have made the following revisions:

  1. add a paragraph in the revised manuscript that acknowledges the limitations and outlines potential future directions for research.

---

## Editorial Decision

published